# Neural Pseudo-Label Optimism for the Bank Loan Problem

**Aldo Pacchiano**[*]  **Shaun Singh**[*]  **Edward Chou**  **Alexander C. Berg**  **Jakob Foerster**
Microsoft Research    FAIR         FAIR           FAIR              FAIR

## Abstract

We study a class of classification problems best exemplified by the *bank loan* problem, where a lender decides whether or not to issue a loan. The lender only observes whether a customer will repay a loan if the loan is issued to begin with, and thus modeled decisions affect what data is available to the lender for future decisions. As a result, it is possible for the lender's algorithm to "get stuck" with a self-fulfilling model. This model never corrects its false negatives, since it never sees the true label for rejected data, thus accumulating infinite regret. In the case of linear models, this issue can be addressed by adding optimism directly into the model predictions. However, there are few methods that extend to the function approximation case using Deep Neural Networks. We present Pseudo-Label Optimism (PLOT), a conceptually and computationally simple method for this setting applicable to DNNs. PLOT adds an optimistic label to the subset of decision points the current model is deciding on, trains the model on all data so far (including these points along with their optimistic labels), and finally uses the resulting *optimistic* model for decision making. PLOT achieves competitive performance on a set of three challenging benchmark problems, requiring minimal hyperparameter tuning. We also show that PLOT satisfies a logarithmic regret guarantee, under a Lipschitz and logistic mean label model, and under a separability condition on the data.

## 1  Introduction

Binary classification models are used for online decision-making in a wide variety of practical settings. These settings include bank lending [44, 22, 42], criminal recidivism prediction [43, 45, 5], credit card fraud [7, 41], spam detection [20, 39], self-driving motion planning [35, 28], and recommendation systems [36, 10, 18]. In many of these tasks, the model only receives the true labels for examples accepted by the learner. We refer to this class of online learning problem as the *bank loan problem* (BLP), motivated by the following prototypical task. A learner interacts with an online sequence of loan applicants. At the beginning of each time-step the learner receives features describing a loan applicant, from which the learner decides whether or not to issue the requested loan. If the learner decides to issue the loan, the learner, at some point in the future, observes whether or not the applicant repays the loan. If the loan is not issued, no further information is observed by the learner. The process is then repeated in the subsequent time-steps. The learner's objective is to maximize its reward by handing out loans to as many applicants that can repay them and deny loans to applicants unable to pay them back.

The BLP can be viewed as a contextual bandit problem with two actions: accepting/rejecting a loan application. Rejection carries a fixed, known, reward of 0: with no loan issued, there is no dependency on the applicant. If in contrast the learner accepts a loan application, it receives a reward of 1 in case the loan is repaid and suffers a loss of $-1$ if the loan is not repaid. The probabilities of repayment

---

[*]Equal contribution.

35th Conference on Neural Information Processing Systems (NeurIPS 2021).

associated to any individual are not known in advance, thus the learner is required to build an accurate predictive model of these while ensuring not too many loans are handed out to individuals who can't repay them and not too many loans are denied to individuals that can repay. This task can become tricky since the samples used to train the model, which govern the model's future decisions, can suffer from bias as they are the result of *past predictions* from a potentially incompletely trained model. In the BLP setting, a model can get stuck in a *self-fulfilling* false rejection loop, in which the very samples that *could* correct the erroneous model never enter the training data in the first place because they are being rejected by the model.

Existing contextual bandit approaches typically assume a known parametric form on the reward function. With restrictions on the reward function, a variety of methods ([13, 8]) introduce strong theoretical guarantees and empirical results, both in the linear and generalized linear model settings[2]. These methods often make use of end-to-end optimism, incorporating uncertainty about the reward function in both the reward model and decision criteria.

In practice, however, deep neural networks (DNNs) are often used to learn the binary classification model [38, 47], presenting to us a scenario that is vastly richer than the linear and generalized linear model assumptions of many contextual bandit works. In the static setting these methods have achieved effective practical performance, and in the case of [47], theoretical guarantees. A large class of these methods use two components: a feature extractor, and a post-hoc exploration strategy fed by the feature extractor. These methods leave the feature extractor itself open to bias, with the limited post-hoc exploration strategy. Another class of methods incorporate uncertainty into the neural network feature extractor ([32]), building on the vast literature on uncertainty estimation in neural networks.

We introduce an algorithm, PLOT (see Algorithm 1), which explicitly trains DNNs to make optimistic online decisions for the BLP, by incorporating *optimism* in both representation learning and decision making. The intuition behind PLOT's optimistic optimization procedure is as follows:

"If I trained a model with the query point having a positive label, would it predict it to be positive?"

If the answer to this question is yes, PLOT would accept this point. To achieve this, at each time step, the PLOT algorithm re-trains its base model, treating the new candidate points as if they had already been accepted and temporarily adds them to the existing dataset with positive *pseudo-labels*. PLOT's accept and reject decisions are based on the predictions from this optimistically retrained base model. The addition of the fake pseudo-labeled points prevents the emergence of *self-fulfilling false negatives*. In contrast to false *rejections*, any false *accepts* introduced by the pseudo-labels are *self-correcting*. Once the model has (optimistically) accepted enough similar data points and obtained their true, negative label, these will overrule the impact of the optimistic label and result in a reject for novel queries.

While conceptually and computationally simple, we empirically show that PLOT obtains competitive performance across a set of 3 different benchmark problems in the BLP domain. With minimal hyperparameter tuning, it matches or outperforms greedy, $\epsilon$-greedy (with a decaying schedule) and state-of-the-art methods from the literature such as NeuralUCB[47]. Furthermore, our analysis shows that PLOT is 3-5 times more likely to *accept* a data point that the current model rejects if the data point is indeed a *true accept*, compared to a *true reject*.

## 1.1 Related Work

**Contextual Bandits** As we formally describe in Section 2, the BLP can be formulated as a contextual bandit. Perhaps the most related setting the BLP in the contextual bandits literature is the work of [30]. In their paper the authors study the loan problem in the presence of offline data. In contrast with our definition for the BLP, their formulation is not concerned with the online decision making component of the BLP, the main point of interest in our formulation. Furthermore, their setting is also concerned with studying ways to satisfy fairness constraints, an aspect of the loan problem that we consider completely orthogonal to our work. Other recent works have forayed into the analysis and deployment of bandit algorithms in the setting of function approximation. Most notably the

---

[2]In a linear model the expected response of a point $\mathbf{x}$ satisfies $\bar{y} = \mathbf{x}^\top \boldsymbol{\theta}$, whereas in a generalized linear model the expected response satisfies $\bar{y} = \mu(\mathbf{x}^\top \boldsymbol{\theta})$ for some non-linearity $\mu$, typically assumed to be the logistic function.

authors of [38] conduct an extensive empirical evaluation of existing bandit algorithms on public datasets. A more recent line of work has focused on devising ways to add optimism to the predictions of neural network models. Methods such as Neural UCB and Shallow Neural UCB [47, 46] are designed to add an optimistic bonus to the model predictions, that is a function of the representation layers of the model. Their theoretical analysis is inspired by insights gained from Neural Tangent Kernel (NTK) theory. Other recent works in the contextual bandit literature such as [14] have started to pose the question of how to extend theoretically valid guarantees into the function approximation scenario, so far with limited success [40]. A fundamental component of our work that is markedly different from previous approaches to is to explicitly encourage optimism throughout representation learning, rather than post-hoc exploration on top of a non-optimistic representation.

**Learning with Abstention**   The literature on learning with abstention shares many features with our setting. In this literature, an online learner can choose to abstain from prediction for a fixed cost, rather than incurring arbitrary regret[9]. In our setting, a rejected point always receives a constant reward, similar to learning with abstention. However, here, regret in the BLP is measured against the potential reward, rather than against a fixed cost. Although the BLP itself does not naturally admit abstention, the extension of PLOT to abstention setting is an interesting future problem.

**Repeated Loss Minimization**   A closely related problem setting to the BLP (see Section 2) is Repeated Loss Minimization. Previous works [17] have studied the problem of repeated classification settings where the acceptance or rejection decision produces a change in the underlying distribution of individuals faced by the decision maker. In their work, the distributional shift induced by the learner's decisions is assumed to be intrinsic to the dynamics of the world. This line of work has recently garnered a flurry of attention and inspired the formalization of different problem domains such as strategic classification [15] and performative prediction [37, 31]. A common theme in these works is the necessity of thinking strategically about the learner's actions and how these may affect its future decisions as a result of the world reacting to them. In this paper we focus on a different set of problems encountered by decision makers when faced with the BLP in the presence of a reward function. We do not treat the world as a strategic decision maker, instead we treat the distribution of data points presented to the learner as fixed, and focus on understanding the effects that making online decisions can have on the future accuracy and reward experienced by an agent engaged in repeated classification. The main goal in this setting is to devise methods that allow the learner to get trapped in false rejection or false acceptance cycles that may compromise its reward. Thus, the learner's task is not contingent on the arbitrariness of the world, but on a precise understanding of its own knowledge of the world.

**Learning with partial feedback**   In [39], the authors study the one-sided feedback setting for the application of email spam filtering and show that the approach of [19] was less effective than a simple greedy strategy. The one-sided feedback setting has in common with our definition of the BLP the assumption that an estimator of the instantaneous regret is only available in the presence of an accept decision. The main difference between the setting of [19] and ours is that the BLP is defined in the presence of possibly noisy labels. Moreover our algorithm PLOT can be used with powerful function approximators, a setting that goes beyond the simple label generating functions studied in [19]. In a related work [4] considers the problem of one-sided learning in the group-based fairness context with the goal of satisfying equal opportunity [16] at every time-step. They consider convex combinations over a finite set of classifiers and arrive at a solution which is a randomized mixture of at most two of these classifiers. Moving beyond the single player one-sided feedback problem [6] studies a setting which generalizes the one-sided feedback, called *partial monitoring*, through considering repeated two-player games in which the player receives a feedback generated by the combined choice of the player and the environment, proposing a randomized solution. [2] provides a classification of such two-player games in terms of the regret rates attained and [3] study a variant of the problem with side information.

## 2   Setting

We formally define the bank loan problem (BLP) as a sequential contextual binary decision problem with conditional labels, where the labels are only observed when datapoints are accepted. In this setting, a decision maker and a data generator interact through a series of time-steps, which we index

by time $t$. At the beginning of every time-step $t$, the decision maker receives a data point $\mathbf{x}_t \in \mathbb{R}^d$ and has to decide whether to accept or reject this point. The label $y_t \in \{0, 1\}$, is only observed if the data point is accepted. If the datapoint is rejected the learner collects a reward of zero. If instead the datapoint is accepted the learner collects a reward of 1 if $y_t = 1$ and $-1$ otherwise. Here, we focus our notation and discussion on the setting where only one point is acted upon in each time-step. All definitions below can be extended naturally to the batch scenario, where a learner receives a batch of data points $\mathbf{x}_{t,1}, \cdots, \mathbf{x}_{t,B}$, and sees labels for accepted points $(y_{t,1} \cdots, y_{t,B'}), y_t \in \{0, 1\}$ in each time-step, where $B$ is the size of the batch, and $B'$ is the number of accepted points.

In the BLP, contexts (unlabeled batches of query points) do not have to be IID – their distributions, $\mathcal{P}(\mathbf{x})$, may change adversarially. As a simple example, the bank may only see applicants for loans under \$1000, until some time point t, where the bank sees applicants for larger loans. Although contexts do not have to be IID, we assume that the reward function itself is always stationary – i.e. the conditional distribution of the responses, $\mathcal{P}(y|\mathbf{x})$, is fixed for all $t$. Finally, in the BLP, it is common for rewards to be delayed – e.g. the learner does not observed the reward for its decision at time $t$ until some later time, $t'$. In this work, we assume that rewards are received immediately, as a wide body of work exists for adapting on-line learning algorithms to delayed rewards [29].

**Reward**   The learner's objective is to maximize its cumulative accuracy, or the number of correct decisions it has made during training, as in [21]. In our model, if the learner accepts the datapoint, $x_t$, it receives a reward of $2y_t - 1$, where $y_t$ is a binary random variable, while a rejection leads to a reward of $0$. Concretely, in the loan scenario, this reward model corresponds to a lender that makes a unit gain for a repaid loan, incurs a unit loss for an unpaid loan and collects zero reward whenever no loan is given.

**Contextual Bandit Reduction**   The BLP can be easily modeled as a specific type of contextual bandit problem [24], where at the beginning of every time-step $t$ the learner receives a context $\mathbf{x}_t$, and has the choice of selecting one of two actions $\{\text{Accept}, \text{Reject}\}$. It is easy to see this characterization implies an immediate reduction of our problem into a two-action contextual bandit problem, where the payout of one of the actions is known by the learner ($r_t(\text{Reject}, x_t) = 0$). To distinguish the problem we study from a more general contextual bandits setting, we refer to our problem setting as the bank-loan problem (BLP).

## 3   Pseudo-Label Optimism

**Overview**   In this section, we describe PLOT in detail, and provide theoretical guarantees for the method. Recall the discussion from Section 1 where we described the basic principle behind the PLOT Algorithm. At the beginning of each time-step $t$, PLOT retrains its base model by adding the candidate points with positive pseudo-labels into the existing data buffer. The learner then decides whether to accept or reject the candidate points by following the predictions from this optimistically trained model. Although the implementation details of PLOT (see Algorithm 1) are a bit more involved than this, the basic operating principle behind the algorithm remains rooted in this very simple idea.

Primarily, PLOT aims to provide similar guarantees as the existing contextual bandit literature, generalized to the function approximation regime. To do so, we rely on the following realizability assumption for the underlying neural model and the distribution of the labels:

**Assumption 1** (Labels generation). *We assume the labels $y_t \in \{0, 1\}$ are generated according to the following model:*

$$y_t = \begin{cases} 1 & \text{with probability } \mu(f_{\theta_\star}(\mathbf{x}_t)) \\ 0 & \text{o.w.} \end{cases} \tag{1}$$

*For some function $f_{\boldsymbol{\theta}_\star} : \mathbb{R}^d \to \mathbb{R}$ parameterized by $\boldsymbol{\theta}_\star \in \Theta$ and where $\mu(z) = \frac{\exp(z)}{1+\exp(z)}$ is the logistic link function. We denote the function class parameterized by $\boldsymbol{\theta} \in \Theta$ as $\mathcal{F}_\Theta = \{f_{\boldsymbol{\theta}} \text{ s.t. } \boldsymbol{\theta} \in \Theta\}$.*

For simplicity, we discuss PLOT under the assumption that $\mathcal{F}_\Theta$ is a parametric class. Our main theoretical results of Theorem 1 hold when this parameterization is rich enough to encompass the set of all constant functions.

In PLOT the learner's decision at time $t$ is parameterized by parameters, $\boldsymbol{\theta}_t$ and a function $f_{\boldsymbol{\theta}_t}$ and takes the form:

$$\text{If } f_{\boldsymbol{\theta}_t}(\boldsymbol{x}_t) \geq 0 \text{ accept}$$

We call the function $f_{\boldsymbol{\theta}_t}$ the learner's model. We denote by $a_t \in \{0, 1\}$ the indicator of whether the learner has decided to accept (1) or reject (0) data point $\mathbf{x}_t$. We measure the performance of a given decision making procedure by its pseudo-regret[3]:

$$\mathcal{R}(t) = \sum_{\ell=1}^{t} \max(0, 2\mu(f_{\boldsymbol{\theta}_\star}(\mathbf{x}_\ell)) - 1) - a_\ell(2\mu(f_{\boldsymbol{\theta}_\star}(\mathbf{x}_\ell)) - 1)$$

For all $t$ we denote the pseudo-reward received at time $t$ as $r_t = a_t(2\mu(f_{\boldsymbol{\theta}_\star}(\boldsymbol{x}_t)) - 1)$. The optimal pseudo-reward at time $t$ equals 0 if $f_{\boldsymbol{\theta}_\star}(\boldsymbol{x}_t) \leq 0$ and $2\mu(f_{\boldsymbol{\theta}_\star}(\boldsymbol{x}_t)) - 1$ otherwise. Minimizing regret is a standard objective in the online learning and bandits literature (see [25]). As a consequence of Assumption 1, the *optimal* reward maximizing decision rule equals the true model $f_{\boldsymbol{\theta}_\star}$.

In order to show theoretical guarantees for our setting we will work with the following realizability assumption.

**Assumption 2** (Neural Realizability). *There exists an $L-$Lipschitz function $f_{\boldsymbol{\theta}_\star} : \mathbb{R}^d \to \mathbb{R}$ such that for all $\mathbf{x}_t \in \mathbb{R}^d$ :*

$$y_t = \begin{cases} 1 & \text{with probability } \mu(f_{\boldsymbol{\theta}_\star}(\mathbf{x}_t)) \\ 0 & \text{o.w.} \end{cases} \tag{2}$$

Recall that in our setting the learner interacts with the environment in a series of time-steps $t = 1, \ldots T$, observing points $\{\mathbf{x}_t\}_{t=1}^{T}$ and labels $y_t$ only during those timesteps when $a_t = 1$. The learner also receives an expected reward of $a_t(2\mu(f_{\boldsymbol{\theta}_\star}(\mathbf{x}_t)) - 1)$, a quantity for which the learner only has access to an unbiased estimator. Whenever Assumption 2 holds, a natural way of computing an estimator $\widehat{\boldsymbol{\theta}}_t$ of $\boldsymbol{\theta}_\star$ is via maximum-likelihood estimation. If we denote by $\mathcal{D}_t = \{(\mathbf{x}_\ell, y_\ell) \text{ s.t.} a_\ell = 1 \text{ and } \ell \leq t - 1\}$ as the dataset of accepted points up to the learners decision at time $t - 1$, the regularized log-likelihood (or negative cross-entropy loss) can be written as,

$$\mathcal{L}^\lambda(\boldsymbol{\theta}|\mathcal{D}_t) = \sum_{(\mathbf{x}, y) \in \mathcal{D}_t} -y \log(\mu(f_{\boldsymbol{\theta}}(\boldsymbol{x}))) - (1 - y) \log(1 - \mu(f_{\boldsymbol{\theta}}(\mathbf{x})) + \frac{\lambda}{2}\|\boldsymbol{\theta}\|_2^2$$

**The Realizable Linear Setting** If $f_{\boldsymbol{\theta}}(\boldsymbol{x}) = \boldsymbol{x}^\top\boldsymbol{\theta}$, the BLP can be reduced to a generalized linear contextual bandit problem (see [13]) via the following reduction. At time $t$, the learner observes a context of the form $\{\mathbf{0}, \boldsymbol{x}_t\}$. In this case, the payoff corresponding to a Reject decision can be realized by action $\mathbf{0}$ for all models in $\mathcal{F}_\Theta$.

Unfortunately, this reduction does not immediately apply to the neural realizable setting. In the neural setting, there may not exist a vector $\mathbf{z} \in \mathbb{R}^d$ with which to model the payoff of the Reject action known a priori to satisfy $f_{\boldsymbol{\theta}_\star}(\mathbf{z}) = 0$. Stated differently, there may not exist an easy way to represent the bank loan problem as a natural instance of a two action contextual bandit problem with the payoffs fully specified by the neural function class at hand. We can get around this issue here, because in the BLP it is enough to compare the model's prediction with the neutral probability $1/2$. Although we make Assumption 2 for the purpose of analyzing and explaining our algorithms, in practice it is not necessary that this assumption holds.

Just as in the case of generalized linear contextual bandits, utilizing the model given by $f_{\hat{\boldsymbol{\theta}}_t}$ may lead to catastrophic under estimation of the true response $\mu(f_{\boldsymbol{\theta}_\star}(\mathbf{x}_t))$ for any query point $\mathbf{x}_t$. The core of PLOT is a method to avoid the emergence of self-fulfilling negative predictions. We do so using a form of implicit optimism, resulting directly from the optimization of a new loss function, which we call the optimistic pseudo-label loss.

---

[3]The prefix pseudo in the naming of the pseudo-regret is a common moniker to indicate the reward considered is a conditional expectation. It has no relation to the pseudo-label optimism of our methods.

**Definition 1** (Optimistic Pseudo-Label Loss). *Let $\mathcal{D} = \{(\mathbf{x}_\ell, y_\ell)\}_{\ell=1}^N$ be a dataset consisting of labeled datapoints $\mathbf{x}_\ell \in \mathbb{R}^d$ and responses $y_\ell \in \{0, 1\}$ and let $\mathcal{B} = \{\mathbf{x}^{(j)}\}_{j=1}^B \subset \mathbb{R}^d$ be a dataset of unlabeled data points. We define the optimistic pseudo-label loss of $(\mathcal{D}, \mathcal{B})$ as,*

$$\mathcal{L}^{\mathcal{C}}(\boldsymbol{\theta}|\mathcal{D}, \mathcal{B}, W, R) = \mathcal{L}^{\lambda}(\boldsymbol{\theta}|\mathcal{D} \cup \{(\mathbf{x}, 1)\, for\, \mathbf{x} \in \mathcal{B}\})$$

$$= \underbrace{\sum_{(\mathbf{x}, y) \in \mathcal{D}(R, \mathcal{B})} -y \log\left(\mu(f_{\boldsymbol{\theta}}(\mathbf{x}))\right) - (1-y)\log(1 - \mu(f_{\boldsymbol{\theta}}(\mathbf{x}))) +}_{\text{cross-entropy loss}}$$

$$\underbrace{W \sum_{\boldsymbol{x} \in \mathcal{B}} \log(\mu(f_{\boldsymbol{\theta}}(\mathbf{x}))}_{\text{pseudo-label loss}} + \frac{\lambda \|\boldsymbol{\theta}\|^2}{2}$$

*Where $W > 0$ is a weighting factor, $R > 0$ is the 'focus' radius and $\mathcal{D}(R, \mathcal{B}) = \{(\mathbf{x}, y) \in \mathcal{D}\ s.t.\ \exists \mathbf{x}' \in \mathcal{B}\ with\ \|\mathbf{x} - \mathbf{x}'\| \leq R\}$.*

Let's take a closer look at the optimistic pseudo-label loss. Given any pair of labeled-unlabeled datasets $(\mathcal{D}, \mathcal{B})$, optimizing for the optimistic loss $\mathcal{L}^{\mathcal{C}}(\theta|\mathcal{D}, \mathcal{B}, W)$ corresponds to minimizing the cross-entropy of a dataset of pairs $(\mathbf{x}, y)$ of the form $(\mathbf{x}, y) \in \mathcal{B}$ or $(\mathbf{x}, 1)$ such that $\mathbf{x} \in \mathcal{B}$. In other words, the minimizer of $\mathcal{L}^{\mathcal{C}}(\boldsymbol{\theta}|\mathcal{D}, \mathcal{B}, W)$ aims to satisfy two objectives:

1. Minimize error on the labeled data
2. Maximize the likelihood of a positive label for the unlabeled points in $\mathcal{B}$.

The model $\widehat{\theta}^{\mathcal{C}}$ resulting from minimizing $\mathcal{L}^{\mathcal{C}}$ will therefore strive to be optimistic over $\mathcal{B}$ while keeping a low loss value, and consequently a high accuracy (when Assumption 2 holds) over the true labels of the points in $\mathcal{D}$. We note that if $\mathcal{D}$ is much larger than $W\mathcal{B}$ (the weighted size of $\mathcal{B}$), optimizing $\mathcal{L}^{\mathcal{C}}$ will favor models that are accurate over $\mathcal{D}$ instead of optimistic over $\mathcal{B}$. Whenever $|\mathcal{D}| \ll W|\mathcal{B}|$, the opposite is true.

**PLOT Algorithm**  Based on these insights, we design Pseudo-Labels for Optimism (PLOT), an algorithm that utilizes the optimistic pseudo-label loss to inject the appropriate amount of optimism into the learner's decisions: high for points that have not been seen much during training, and low for those points whose acceptance may cause a catastrophic loss increase over the points accepted by the learner so far.

---

**Algorithm 1** Pseudo-Labels for Optimism (PLOT)

---

**Input:** $\epsilon-$greedy exploration parameter, weight schedule $\{W_t\}_{t=1}^\infty$, radius $R$
Initialize accepted dataset $\mathcal{D}_1 = \emptyset$
**for** $t = 1, \cdots T$ **do**

    1. Observe batch $\mathcal{B}_t = \{\mathbf{x}_t^{(j)}\}_{j=1}^B$ and sample $Q_t = \{q_j\}_{j=1}^B$ such that $q_j \overset{i.i.d.}{\sim} \text{Ber}(\epsilon)$
    2. Build the MLE estimator,
$$\widehat{\boldsymbol{\theta}}_t = \min_{\boldsymbol{\theta}} \mathcal{L}_t(\boldsymbol{\theta}|\mathcal{D}_t)$$

    3. Compute the pseudo-label filtered batch $\widetilde{\mathcal{B}}_t = \{(\mathbf{x}_t^{(j)}, 1)\ \text{s.t.}\ f_{\widehat{\boldsymbol{\theta}}_t}(\mathbf{x}_t^{(j)}) < 0\ \text{and}\ q_j = 1\}$.
    4. Calculate the minimizer of the empirical optimistic pseudo-label loss,
$$\widehat{\boldsymbol{\theta}}_t^{\mathcal{C}} = \min_{\boldsymbol{\theta}} \mathcal{L}_t^{\mathcal{C}}(\boldsymbol{\theta}|\mathcal{D}_t, \widetilde{\mathcal{B}}_t, W_t, R),$$

    3. For all $\mathbf{x}^{(j)} \in \mathcal{B}_t$ compute acceptance decision via $a_t^{(j)} = \begin{cases} 1 & \text{if } f_{\widehat{\boldsymbol{\theta}}_t^{\mathcal{C}}}(\mathbf{x}_t^{(j)}) \geq 0 \\ 0 & \text{o.w.} \end{cases}$

    4. Update $\mathcal{D}_{t+1} \leftarrow \mathcal{D}_t \cup \{(\mathbf{x}_t^{(j)}, y_t^{(j)})\}_{j \in \{1, \cdots, B\}\ \text{s.t.}\ a_t^{(j)} = 1}$.

**end**

---

During the very first time-step ($t = 1$), PLOT accepts all the points in $\mathcal{B}_t$. In subsequent time-steps PLOT makes use of a dual $\epsilon-$greedy and MLE-greedy filtering subroutine to find a subset of the

current batch composed of those points that are *both* currently being predicted as rejects by the MLE estimator and have been selected by the $\epsilon-$greedy schedule (see step 3 of PLOT).

This dual filtering mechanism ensures that only a small proportion (based on the $\epsilon$) of the datapoints are ever considered to be included into the empirical optimistic pseudo-label loss. The MLE filtering mechanism further ensures that not all the points selected by $\epsilon-$greedy are further investigated, but only those that are currently being rejected by the MLE model. This has the effect of preventing the pseudo-label filtered batch from growing too large.

As we have mentioned above, the relative sizes of the labeled and unlabeled batches has an effect on the degree of optimism the algorithm will inject into its predictions. As the size of the collected dataset grows, the inclusion of $\widetilde{\mathcal{B}}_t$, has less and less effect on $\widehat{\theta}_t^{\mathcal{C}}$. In the limit, once the dataset is sufficiently large and accurate information can be inferred about the true labels, the inclusion of $\widetilde{\mathcal{B}}_t$ into the pseudo-label loss has vanishing effect. The later has the beneficial effect of making false positive rate decrease with $t$.

The following guarantee shows that in the case of separable data satisfying Assumption 2, the PLOT Algorithm initialized with the right parameters $R, \{W_t\}_{t=1}^\infty$ satisfies a logarithmic regret guarantee.

**Theorem 1.** *Let $\mathcal{P}$ be a distribution over data point, label pairs $(\mathbf{x}, y)$ satisfying*

1. *All $\mathbf{x} \in \text{supp}(\mathcal{P})$ are bounded $\|\mathbf{x}\| \leq B$.*

2. *The conditional distributions of the labels $y$ satisfy the data generating Assumption 2 with $\mathcal{F}$ a class of $L-$Lipschitz functions containing all constant functions.*

3. *$|f_{\boldsymbol{\theta}_\star}(\mathbf{x})| \geq \tau > 0$ holds for all $\mathbf{x} \in \text{supp}(\mathcal{P})$.*

*Let the marginal distribution of $\mathcal{P}$ over points $\mathbf{x}$ be $\mathcal{P}_{\mathcal{X}}$ and let's assume the PLOT algorithm will be used in the presence of i.i.d. data such that $\mathbf{x}_t \sim \mathcal{P}_{\mathcal{X}}$ independently for all $t \in \mathbb{N}$. Define $A_t = \sum_{\ell=1}^{t-1} \mathbf{1}\{\mathbf{x}_\ell \in B(\mathbf{x}_t, R)\}$ and $D_t = \sum_{\ell=1}^{t-1} y_\ell \mathbf{1}\{\mathbf{x}_\ell \in B(\mathbf{x}_t, R)\}$ where $B(\mathbf{x}, R)$ corresponds to the $\| \cdot \|_2$ ball centered at $\mathbf{x}$ and radius $R$. Let $\delta' \in (0, 1)$. If $R = \frac{\tau^2}{128L}$, $W_t = \max\left(4\sqrt{t \ln\left(\frac{6t^2 \ln t}{\delta'}\right)}, \frac{\left(\frac{\mu(\tau)}{2} + \frac{1}{4}\right) A_t - D_t}{\frac{3}{4} - \frac{\mu(\tau)}{2}}\right)$ and $\epsilon = 1$, the PLOT Algorithm with batch size 1 satisfies for all $t \in \mathbb{N}$ simultaneously,*

$$\mathcal{R}(t) \leq \widetilde{\mathcal{O}}\left(\frac{1}{\tau p_{\mathcal{X}}} \ln\left(\frac{1}{\delta'}\right)\right)$$

*With probability at least $1 - \delta'$, where $p_{\mathcal{X}}$ is a parameter that only depends on the geometry of $\mathcal{P}_{\mathcal{X}}$ and $\widetilde{\mathcal{O}}$ hides logarithmic factors in $\tau$ and $p_{\mathcal{X}}$.*

The proof can be found in Appendix **??**. Although the guarantees of Theorem 1 require knowledge of $\tau$, in practice this requirement can easily be alleviated by using any of a variety of Model Selection approaches such as in [34, 11, 1, 27, 33], at the price of a slightly worse regret rate. In the following section we conduct extensive empirical studies of PLOT and demonstrate competitive finite time regret on a variety of public classification datasets.

## 4   Experimental Results

**Experiment Design and Methods**   We evaluate the performance of PLOT[4] on three binary classification problems adapted to the BLP setting. In time-step $t$, the algorithm observes context $\mathbf{x}_t \in \mathbb{R}^d$, and classifies the point, a.k.a accepts/rejects the point. If the point is accepted, a reward of one is given if that point is from the target class, and minus one otherwise. If the point is rejected, the reward is zero. We focus on two datasets from the UCI Collection [12], the Adult dataset and the Bank dataset. Additionally we make use of MNIST [26] (d=784). The Adult dataset is defined as a binary classification problem, where the positive class has income > \$50k. The Bank dataset is also binary, with the positive class as a successful marketing conversion. On MNIST, we convert the multi-class problem to binary classification by taking the positive class to be images of the digit 5,

---

[4]Google Colab: shorturl.at/pzDY7

and treating all other images as negatives. Our main metric of interest is regret, measured against a baseline model trained on the entire dataset. The baseline model is used instead of the true label, as many methods cannot achieve oracle accuracy on real-world problems even with access to the entire dataset.

We focus on comparing our method to other neural algorithms, as prior papers [38], [23], [47] generally find neural models to have the best performance on these datasets. In particular, we focus on NeuralUCB[47] as a strong benchmark method. We perform a grid search over a few values of the hyperparameter of NeuralUCB, considering {0.1, 1, 4, 10} and report results from the best value. We also consider greedy and $\epsilon$-greedy methods. For $\epsilon$-greedy, we follow [23], and give the method an unfair advantage, i.e. we use a decayed schedule, dropping to 0.1% exploration by T=2000. Otherwise, the performance is too poor to plot.

In our experiments, we set the PLOT weight parameter to 1, equivalent to simply adding the pseudo-label point to the dataset. We set the PLOT radius parameter to $\infty$, thus including all prior observed points in the training dataset. Although our regret guarantees require problem-dependent settings of these two parameters, PLOT achieves strong performance with these simple and intuitive settings, without sweeping.

For computational efficiency, we run our method on batches of data, with batch size $n = 32$. We average results over 5 runs, running for a horizon of $t = 2000$ time-steps. Our dataset consists of the points accepted by the algorithm, for which we have the true labels. We report results for a two-layer, 40-node, fully-connected neural network. At each timestep, we train this neural network on the above data, for a fixed number of steps. Then, a second neural network is cloned from those weights. A new dataset with pseudo-label data and historical data is constructed, and the second neural network is trained on that dataset for the same number of steps. This allows us to keep a continuously trained model which only sees true labels. The pseudo-label model only ever sees one batch of pseudo-labels. Each experiment runs on a single Nvidia Pascal GPU, and replicated experiments, distinct datasets, and methods can be run in parallel, depending on GPU availability.

**Analysis of Results** In the top row of Figure 1, we provide cumulative regret plots for the above datasets and methods. Our method's cumulative regret is consistently competitive with other methods, and outperforms on MNIST. In addition, the variance of our method is much lower than that of NeuralUCB and Greedy, showing very consistent performance across the five experiments.

The bottom row of Figure 1 provides a breakdown of the decisions made by our model. As described in Section 3, on average the pseudo-label model only acts on $\epsilon$-percent of points classified as negative by the base model. We provide the cumulative probability of acceptance of true positive and true negative points acted on by the pseudo-label model. As the base model improves, the pseudo-label model receives fewer false positives, and becomes more confident in supporting rejections from the base model. To differentiate this decaying process from the pseudo-label learning, we highlight the significant gap between the probability of accepting positives and the probability of accepting negatives in our method. This shows that the PLOT method is not simply performing a decayed-$\epsilon$ strategy, but rather learning for which datapoints to inject optimism into the base classifier.

**PLOT in action.** We illustrate the workings of the PLOT algorithm by testing it on a simple XOR dataset. In Figure 2 we illustrate the evolution of the model's decision boundary in the presence of pseudo-label optimism. On the top left panel of Figure 2 we plot 300 samples from the XOR dataset. There are four clusters in the XOR dataset. Each of these is produced by sampling a multivariate normal with isotropic covariance with a diagonal value of $0.5$. The cluster centers are set at $(0, 5), (0, 0), (5, -2)$, and $(5, 5)$. All points sampled from the red clusters are classified as $0$ and all points sampled from the black clusters are classified as $1$. Although there are no overlaps in this picture, there is a non-zero probability that a black point may be sampled from deep inside a black region and vice versa.

## 5 Conclusion

We propose PLOT, a novel algorithm that provides end-to-end optimism for the bank loan problem with neural networks. Rather than post-hoc optimism separate from the neural net training, optimism is directly incorporated into the neural net loss function through the addition of optimistic pseudo-

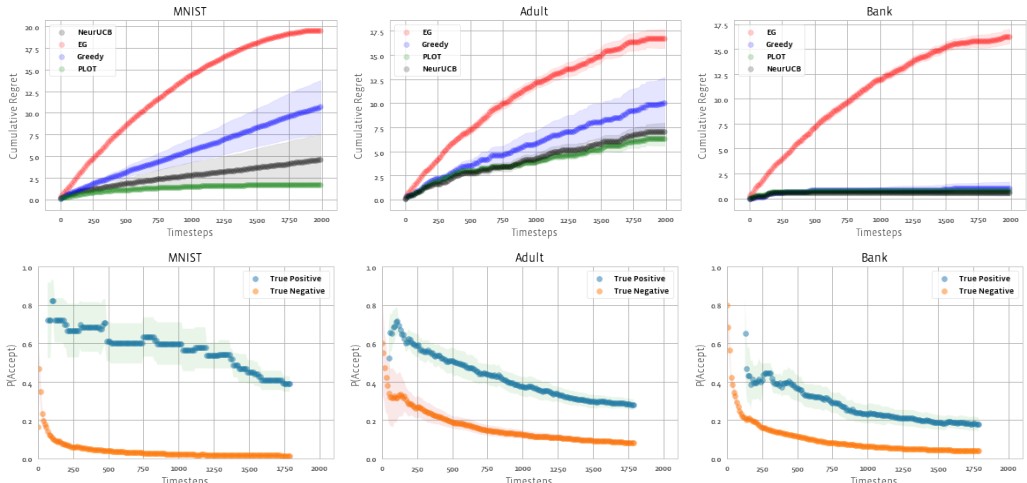

Figure 1: Comparison of PLOT to $\epsilon$-greedy (EG), greedy, and NeuralUCB[47]. Reward and pseudo-label accuracy are reported as a function of the timestep. One standard deviation from the mean, computed across the five experiments, is shaded.

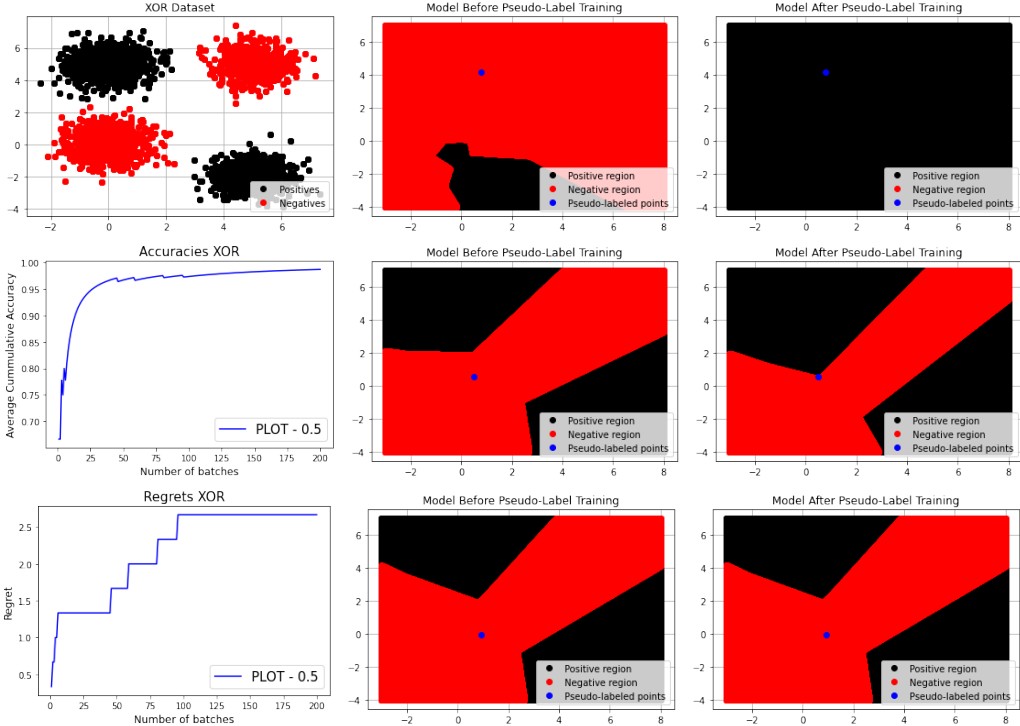

Figure 2: PLOT with parameter $\epsilon = .5$ and batch size equals 3. **Top** model boundary for batch 1. The model's test accuracy after training equals $50\%$ Having seen very little data the model is very sensitive to pseudo-label optimism. **Middle** model boundary for batch 81. The model's test accuracy equals $97.8\%$. The model boundary is pulled towards the pseudo labeled point. **Bottom** model boundary for batch 105. The model's test accuracy equals $99.9\%$. The model training has stabilized. The extra optimism does not change the model boundary.

labels. We provide regret guarantees for `PLOT`, and demonstrate its performance on real-world problems, where its performance illustrates the value of end-to-end optimism.

Our current analysis and deployment of `PLOT` is focused on the bank loan problem, with binary actions, where pseudo-label semantics are most clear. Due to the deep connections with active learning and binary contextual bandits, extending this work to larger action spaces is an interesting future direction.

Although the BLP is naturally modeled with delayed feedback, PLOT assumes that rewards are received immediately, as a wide body of work exists for adapting on-line learning algorithms to delayed rewards [29]. Contexts (unlabeled batches of query points) do not have to be IID – their distributions, $\mathcal{P}(\mathbf{x})$, may change adversarially. Handling this type of shift is a key component of PLOT's approach. Optimism is essential to avoiding feedback loops in online learning algorithms, with significant implications for the fairness literature. We presented regret analyses here, which we hope can lay the foundation for future work on the analysis of optimism in the fairness literature.

## 6  Statement of Broader Impact

Explicitly incorporating optimism into neural representation learning is key to ensuring optimal exploration in the *bank loan* problem. Other methods for exploration run the risk of feature blindness, where a neural network loses its uncertainty over certain features. When a representation learning method falls victim to this, additional optimism is insufficient to ensure exploration. This has ramifications for fairness and safe decision making. We believe that explicit optimism is a key step forward for safe and fair decision making.

However, we do want to provide caution around our method's limitations. Our regret guarantees and empirical results assume I.I.D data, and may not prevent representation collapse in non-stationary and adversarial settings. Additionally, although our empirical results show strong performance in the non-separable setting, our regret guarantees only hold uniformly in the separable setting.

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
