# A  Why Optimism?

In this section we describe the common proof template behind the principle of optimism in Stochastic Bandit problems. We illustrate this in the setting of binary classification that we work with.

As we mentioned in Section 2, we work with decision rules based on producing at time $t$ a model $\boldsymbol{\theta}_t$ that is used to make a prediction of the form $\mu(f_{\boldsymbol{\theta}_t}(\mathbf{x}_t))$ of the probability that point $\mathbf{x}_t$ should be accepted. If $f_{\boldsymbol{\theta}_t}(\mathbf{x}_t) \geq 0$, point $\mathbf{x}_t$ will be accepted and its label $y_t$ observed, whereas if $f_{\boldsymbol{\theta}_t}(\mathbf{x}_t) < 0$, the point will be discarded and the label will remain unseen. Here we define an optimistic algorithm in this setting:

**Definition 2** (Optimistic algorithm). *We say an algorithm is optimistic for this setting if the models selected at all times $t$ satisfy $f_{\boldsymbol{\theta}_t}(\mathbf{x}_t) \geq f_{\boldsymbol{\theta}_\star}(\mathbf{x}_t)$ for all $t$.*

We now show the regret of any optimistic algorithm can be upper bounded by the model's estimation error,

$$
\mathcal{R}(t) = \sum_{\ell=1}^{t} \max(0, 2\mu(f_{\boldsymbol{\theta}_\star}(\mathbf{x}_t)) - 1) - a_t(2\mu(f_{\boldsymbol{\theta}_\star}(\mathbf{x}_t)) - 1)
$$

$$
\overset{(i)}{\leq} \sum_{\ell=1}^{t} 2a_t \left( \mu(f_{\boldsymbol{\theta}_t}(\mathbf{x}_t)) - \mu(f_{\boldsymbol{\theta}_\star}(\mathbf{x}_t)) \right)
$$

Let's see why inequality $(i)$ holds. Notice that for any optimistic model, the false negative rate must be zero. Rejection of a point $\mathbf{x}_t$ may occur only for points that are truly negative. This implies the instantaneous regret satisfies

$$
\max(0, 2\mu(f_{\boldsymbol{\theta}_\star}(\mathbf{x}_t))-1)-a_t(2\mu(f_{\boldsymbol{\theta}_\star}(\mathbf{x}_t))-1) = a_t \left( \max(0, 2\mu(f_{\boldsymbol{\theta}_\star}(\mathbf{x}_t)) - 1) - 2\mu(f_{\boldsymbol{\theta}_\star}(\mathbf{x}_t)) + 1 \right).
$$

By definition $a_t = 1$ only when $f_{\boldsymbol{\theta}_t}(\mathbf{x}_t) \geq 0$. This observation plus the optimistic nature of the models $\{\boldsymbol{\theta}_t\}_t$ implies that $\max(0, 2\mu(f_{\boldsymbol{\theta}_\star}(\mathbf{x}_t)) - 1) \leq 2\mu(f_{\boldsymbol{\theta}_t}(\mathbf{x}_t)) - 1$ and thus inequality $(i)$.

As a consequence of this discussion we can conclude that in order to control the regret of an optimistic algorithm, it is enough to control its estimation error. In other words, finding a model that overestimates the response is not sufficient, the models' error must converge as well.

# B  PLOT **Theory - Proof of Theorem 1**

In this section we prove the results stated in Theorem 1. The following property of the logistic function will prove useful.

**Remark 1.** *The logistic function $\mu$ is $1/4$ Lipschitz.*

Throughout the discussion we will make use of the notation $B(\mathbf{x}, R)$ to denote the $\|\cdot\|_2$ ball of radius $R$ centered around point $\mathbf{x}$.

In this section we will make the following assumptions.

**Assumption 3** (Bounded Support $\mathcal{P}_{\mathcal{X}}$). *$\mathcal{P}_{\mathcal{X}}$ has bounded support. All $\mathbf{x} \in \mathbf{supp}(\mathcal{P}_{\mathcal{X}})$ satisfy $\|\mathbf{x}\| \leq B$.*

**Assumption 4** (Lipschitz $\mathcal{F}_{\Theta}$). *The function class $\mathcal{F}_{\Theta}$ is $L-$Lipschitz and contains all constant functions ( $f_{\boldsymbol{\theta}}$ such that $f_{\boldsymbol{\theta}}(\mathbf{x}) = c$ for $c \in [-2, 2]$).*

**Assumption 5** ($\tau-$Gap). *For all $x \in \mathbf{supp}(\mathcal{P}_{\mathcal{X}})$, the values $f_{\boldsymbol{\theta}_\star}(x)$ are bounded away from zero.*

$$
|f_{\boldsymbol{\theta}_\star}(\boldsymbol{x})| \geq \tau > 0.
$$

*where $\tau \in (0, 1)$.*

The following supporting result regarding the logistic function will prove useful.

**Lemma 1.** *For $x \in (0, 1)$, the logistic function satisfies, $\frac{1}{2} + cx \leq \mu(x) \leq \frac{1}{2} + x$ where $c = \frac{e}{(1+e)^2}$ and $e = \exp(1)$.*

*Proof.* The derivative of $\mu$ satisfies $\mu'(x) = (1 - \mu(x))\mu(x)$ and is a decreasing function in the interval $(0, 1)$ with a minimum value of $\frac{e}{(1+e)^2}$.

Consider the function $g(x) = \mu(x) - \left(\frac{1}{2} + cx\right)$. It is easy to see that $g(0) = 0$ and that $g'(x) = \mu'(x) - c \geq 0$ for all $x \in (0, 1)$, therefore, $g(x)$ is increasing in the interval $(0, 1)$ and we conclude that $g(x) \geq 0$ for all $x \in (0, 1)$. The result follows:

$$\mu(x) \geq \frac{1}{2} + cx \quad \forall x \in (0, 1).$$

To prove the second direction we consider the function $h(x) = \frac{1}{2} + x - \mu(x)$. Observe that $h'(x) = 1 - (1 - \mu(x))\mu(x)$ and therefore $h'(x) \geq 0$ since $h(0) = 0$ this implies ther result.

$\square$

We will make use of Pinsker's inequality,

**Lemma 2** (Pinsker's inequality). *Let $\mathbb{P}$ and $\mathbb{Q}$ be two distributions defined on the unvierse $U$. Then,*

$$D_{\mathrm{KL}}(\mathbb{P} \parallel \mathbb{Q}) = \frac{1}{2\ln(2)}\|\mathbb{P} - \mathbb{Q}\|_1^2$$

Recall the unregularized and normalized negative cross entropy loss over a dataset $\mathcal{D}_t$ equals,

$$\bar{\mathcal{L}}(\boldsymbol{\theta}|\mathcal{D}_t) = \frac{1}{|\mathcal{D}_t|} \sum_{(\mathbf{x},y) \in \mathcal{D}_t} -y \log\left(\mu(f_{\boldsymbol{\theta}}(\boldsymbol{x}))\right) - (1 - y) \log\left(1 - \mu(f_{\boldsymbol{\theta}}(\mathbf{x}))\right) \tag{3}$$

We can extend this definition to the population level. For any distribution $\mathcal{Q}$ we define the unregularized *normalized* cross entropy loss over $\mathcal{Q}$ whose labels are generated according to a logistic model with parameter $\boldsymbol{\theta}_\star$ as

$$\bar{\mathcal{L}}(\boldsymbol{\theta}|\mathcal{Q}) = \mathbb{E}_{(\mathbf{x},y)\sim\mathcal{Q}} \left[-y \log\left(\mu(f_{\boldsymbol{\theta}}(\boldsymbol{x}))\right) - (1 - y) \log\left(1 - \mu(f_{\boldsymbol{\theta}}(\mathbf{x}))\right)\right]$$
$$= \mathbb{E}_{\mathbf{x}\sim\mathcal{Q}_{\mathbf{x}}} \left[\mathrm{KL}\left(\mu(f_{\boldsymbol{\theta}_\star}(\mathbf{x}) \parallel \mu(f_{\boldsymbol{\theta}}(\mathbf{x}))\right)\right] - \mathbb{E}_{\mathbf{x}\sim\mathcal{Q}_{\mathbf{x}}} \left[\mathrm{H}(\mu(f_{\boldsymbol{\theta}_\star}(\mathbf{x}))\right]$$

As an immediate consequence of the last equality, we see that when $f_{\boldsymbol{\theta}_\star} \in \mathcal{F}$, the vector $\boldsymbol{\theta}_\star$ is a minimizer of the population cross entropy loss. From now on we'll use the notation $\widehat{\mathcal{P}}_t$ to denote the empirical distribution over datapoints given by $\mathcal{D}_t$.

Observe also that if $\boldsymbol{x} \in \mathbf{supp}(\mathcal{P}_{\mathcal{X}})$, for all $\boldsymbol{x}' \in \mathbb{R}^d$ such that $\|\boldsymbol{x} - \boldsymbol{x}'\| \leq \frac{\tau}{2L}$, we have that as a consequence of Assumption 4,

$$|f_{\boldsymbol{\theta}}(\boldsymbol{x}) - f_{\boldsymbol{\theta}}(\boldsymbol{x}')| \leq \frac{\tau}{2}, \quad \forall \boldsymbol{\theta} \in \Theta.$$

and therefore because of Assumption 5,

$$|f_{\boldsymbol{\theta}_\star}(\boldsymbol{x}')| \geq \frac{\tau}{2}.$$

Now let's consider $\boldsymbol{x} \in \mathbf{supp}(\mathcal{P}_{\mathcal{X}})$ such that $f_{\boldsymbol{\theta}}(\boldsymbol{x}) > 0$. By Assumption 5, this implies that $f_{\boldsymbol{\theta}}(\boldsymbol{x}) \geq \tau$. Similarly if $\boldsymbol{x} \in \mathbf{supp}(\mathcal{P}_{\mathcal{X}})$ such that $f_{\boldsymbol{\theta}}(\boldsymbol{x}) < 0$ implies that $f_{\boldsymbol{\theta}}(\boldsymbol{x}) \leq -\tau$.

Let's start by considering the case when $\forall (\mathbf{x}, y), (\mathbf{x}', y') \in \mathcal{D}_t$ satisfy $\|\mathbf{x} - \mathbf{x}'\| \leq \tau^2$. Let's for a moment assume that $f_{\boldsymbol{\theta}_\star}(\mathbf{x}) > 0$ for all $(\mathbf{x}, y) \in \mathcal{D}_t$ and therefore (by Assumption 5) that $f_{\boldsymbol{\theta}_\star}(\mathbf{x}) \geq \tau$. If this is the case, we will assume that

**Lemma 3.** *If $\mathcal{D}_t$ satisfies the following properties,*

1. *$\forall (\mathbf{x}, y), (\mathbf{x}', y') \in \mathcal{D}_t$ it holds that $\|\mathbf{x} - \mathbf{x}'\| \leq \frac{\tau^2}{128L}$.*

2. *There exists $(\tilde{\mathbf{x}}, y) \in \mathcal{D}_t$ such that $f_{\boldsymbol{\theta}_\star}(\mathbf{x}) > 0$.*

3. $\hat{y} = \frac{1}{|\mathcal{D}_t|} \sum_{(\mathbf{x},y) \in \mathcal{D}_t} y \geq \frac{1}{4} + \frac{\mu(\tau)}{2}$.

*Then,*

$$\widehat{\boldsymbol{\theta}}_t = \arg\min_{\theta} \bar{\mathcal{L}}(\boldsymbol{\theta}|\mathcal{D}_t)$$

*Satisfies, $f_{\widehat{\boldsymbol{\theta}}_t}(\mathbf{x}) > 0$ for all $(\mathbf{x},y) \in \mathcal{D}_t$.*

*Proof.* First observe that as a consequence of the $L$-Lipschitzness of $f_{\boldsymbol{\theta}_\star}$ having all points in $\mathcal{D}_t$ be contained within a ball of radius $\frac{\tau^2}{128L}$ implies that for all $(\mathbf{x},y), (\mathbf{x}',y') \in \mathcal{D}_t$ the difference $|\mu(f_{\boldsymbol{\theta}_\star}(\mathbf{x})) - \mu(f_{\boldsymbol{\theta}_\star}(\mathbf{x}'))| \leq \frac{1}{4}|f_{\boldsymbol{\theta}_\star}(\mathbf{x}) - f_{\boldsymbol{\theta}_\star}(\mathbf{x}')| \leq \frac{L}{4}\|\mathbf{x} - \mathbf{x}'\| \leq \frac{L\tau^2}{4 \times 128L} = \frac{\tau^2}{128 \times 4}$. In particular this also implies that $|f_{\boldsymbol{\theta}_\star}(\mathbf{x}) - f_{\boldsymbol{\theta}_\star}(\mathbf{x}')| \leq \frac{\tau^2}{128} \leq \frac{\tau}{128}$. The last inequality holds because $\tau^2 \leq \tau$.

Let $\widetilde{\mathbf{x}}$ be a point in $\mathcal{D}_t$ such that $f_{\boldsymbol{\theta}_\star}(\widetilde{\mathbf{x}}) > 0$. By Assumption 5, $f_{\boldsymbol{\theta}_\star}(\widetilde{\mathbf{x}}) \geq \tau$ and therefore, $\mu(f_{\boldsymbol{\theta}_\star}(\widetilde{\mathbf{x}})) \geq \mu(\tau)$. This implies that for all $(\mathbf{x},y) \in \mathcal{D}_t$, $\mu(f_{\boldsymbol{\theta}_\star}(\mathbf{x})) \geq \mu(\tau) - \frac{\tau}{128 \times 4}$ and that $f_{\boldsymbol{\theta}_\star}(\mathbf{x}) \geq \frac{127\tau}{128}$.

By Lemma 1 $\mu(\tau) \geq \frac{1}{2} + c\tau$ where $c \approx .196$ and therefore $\mu(f_{\boldsymbol{\theta}_\star}(\mathbf{x})) \geq \frac{1}{2} + (c - \frac{1}{128 \times 4})\tau \geq \frac{1}{2} + \frac{4\tau}{25}$ for all $(\mathbf{x},y) \in \mathcal{D}_t$. In other words, all points in $\mathcal{D}_t$ should have true positive average labels with a probability gap value (away from $1/2$) of at least $\frac{4\tau}{25}$.

We will prove this Lemma by exhibiting an $L-$Lipschitz classifier whose loss always lower bounds the loss of any classifier that rejects any of the points. But first, let's consider a classifier parametrized by $\tilde{\boldsymbol{\theta}}$ such that $f_{\tilde{\boldsymbol{\theta}}}(\mathbf{x}) \leq 0$ for some $(\mathbf{x},y) \in \mathcal{D}_t$. If this holds, the radius $\frac{\tau}{128L}$ of $\mathcal{D}_t$ and the $L-$Lipschitzness of the function class imply,

$$f_{\tilde{\boldsymbol{\theta}}}(\mathbf{x}') \leq \frac{\tau}{128}, \forall \mathbf{x}' \in \mathcal{D}_t.$$

And therefore that $\mu(f_{\tilde{\boldsymbol{\theta}}}(\mathbf{x})) \leq \mu(\frac{\tau}{128}) \leq \frac{1}{2} + \frac{\tau}{128}$. Similar to the argument we made for $\boldsymbol{\theta}_\star$ above, Lipschitzness implies,

$$\left|\mu(f_{\tilde{\boldsymbol{\theta}}}(\mathbf{x})) - \mu(f_{\tilde{\boldsymbol{\theta}}}(\mathbf{x}'))\right| \leq \frac{L\tau^2}{4 \times 128L} = \frac{\tau^2}{128 \times 4} \leq \frac{\tau}{128 \times 4} \tag{4}$$

Combining these observations we conclude that

$$\mu(f_{\tilde{\boldsymbol{\theta}}}(\mathbf{x})) < \frac{1}{2} + \frac{\tau}{128} < \frac{1}{2} + \frac{4\tau}{25} \leq \mu(f_{\boldsymbol{\theta}_\star}(\mathbf{x})), \quad \forall \mathbf{x} \in \mathcal{D}_t. \tag{5}$$

Let's now consider $\boldsymbol{\theta}_{\text{const}}$ be a parameter such that $f_{\boldsymbol{\theta}_{\text{constant}}}(\mathbf{x}) = \mu^{-1}(\frac{1}{2} + \frac{487\tau}{6400})$ so that $\mu(f_{\boldsymbol{\theta}_{\text{constant}}}(\mathbf{x})) = \frac{1}{2} + (\frac{4}{25} - \frac{1}{128})/2 = \frac{1}{2} + \frac{487\tau}{6400}$ for all $\mathbf{x} \in \mathcal{D}_t$. This is a constant classifier whose responses lie exactly midway between the lower bounds for the predictions of $\boldsymbol{\theta}_\star$ and $\tilde{\boldsymbol{\theta}}$.

Denote by $\mathcal{D}_t(1) = \{(\mathbf{x},y) \in \mathcal{D}_t \text{ s.t. } y = 1\}$ and $\mathcal{D}_t(0) = \{(\mathbf{x},y) \in \mathcal{D}_t \text{ s.t. } y = 0\}$.

Recall that $\mathcal{L}^0(\boldsymbol{\theta}|\mathcal{D}_t) = \sum_{(\mathbf{x},y) \in \mathcal{D}_t} -y \log\left(\mu(f_{\boldsymbol{\theta}}(\mathbf{x})) - (1-y)\log\left(1 - \mu(f_{\boldsymbol{\theta}}(\mathbf{x})\right)\right.$. Hence,

$$
\begin{aligned}
\mathcal{L}^0(\tilde{\boldsymbol{\theta}}|\mathcal{D}_t) - \mathcal{L}^0(\boldsymbol{\theta}_{\text{constant}}|\mathcal{D}_t) &= \sum_{(\mathbf{x},y) \in \mathcal{D}_t} y \log\left(\frac{\mu(f_{\boldsymbol{\theta}_{\text{constant}}}(\mathbf{x}))}{\mu(f_{\tilde{\boldsymbol{\theta}}}(\mathbf{x}))}\right) + \\
&\quad (1-y)\log\left(\frac{1 - \mu(f_{\boldsymbol{\theta}_{\text{constant}}}(\mathbf{x}))}{1 - \mu(f_{\tilde{\boldsymbol{\theta}}}(\mathbf{x}))}\right) \\
&= \sum_{(\mathbf{x},y) \in \mathcal{D}_t(1)} \log\left(\frac{\mu(f_{\boldsymbol{\theta}_{\text{constant}}}(\mathbf{x}))}{\mu(f_{\tilde{\boldsymbol{\theta}}}(\mathbf{x}))}\right) + \\
&\quad \sum_{(\mathbf{x},y) \in \mathcal{D}_t(0)} \log\left(\frac{1 - \mu(f_{\boldsymbol{\theta}_{\text{constant}}}(\mathbf{x}))}{1 - \mu(f_{\tilde{\boldsymbol{\theta}}}(\mathbf{x}))}\right)
\end{aligned}
$$

By Equations 4, 5,

$$\min_{0 \le z \le \frac{1}{2} + \frac{\tau}{128}} |\mathcal{D}_t(1)| \log\left(\frac{\mu(f_{\boldsymbol{\theta}_{\text{constant}}}(\mathbf{x}))}{z}\right) + |\mathcal{D}_t(0)| \log\left(\frac{1 - \mu(f_{\boldsymbol{\theta}_{\text{constant}}}(\mathbf{x}))}{1 - z + \frac{\tau^2}{512}}\right)$$
$$\le \mathcal{L}^0(\tilde{\boldsymbol{\theta}}|\mathcal{D}_t) - \mathcal{L}^0(\boldsymbol{\theta}_{\text{constant}}|\mathcal{D}_t)$$

Notice that $\mu(f_{\boldsymbol{\theta}_{\text{constant}}}(\mathbf{x})) = \frac{1}{2} + \frac{487\tau}{6400} > \frac{1}{2} + \frac{\tau}{128} + \frac{\tau^2}{512}$ and therefore $\log\left(\frac{1 - \mu(f_{\boldsymbol{\theta}_{\text{constant}}}(\mathbf{x}))}{1 - z + \frac{\tau^2}{512}}\right) \le 0$ for all $z \le \frac{1}{2} + \frac{\tau}{128}$.

Recall that by Assumption 5, the gap $\tau \in (0, 1)$ and therefore by Lemma 1, $\mu(\tau) \ge \frac{1}{2} + c\tau$ where $c \approx .196$. Let's try showing that $\frac{\mathcal{L}^0(\tilde{\boldsymbol{\theta}}|\mathcal{D}_t) - \mathcal{L}^0(\boldsymbol{\theta}_{\text{constant}}|\mathcal{D}_t)}{|\mathcal{D}_t|} > 0$. Since by assumption $\frac{|\mathcal{D}_t(1)|}{|\mathcal{D}_t|} \ge \frac{1}{4} + \frac{\mu(\tau)}{2} \ge \frac{1}{2} + \frac{c\tau}{2} \ge \frac{1}{2} + \frac{49\tau}{500}$, this statement holds if

$$\min_{0 \le z \le \frac{1}{2} + \frac{\tau}{128}} \left(\frac{1}{2} + \frac{49\tau}{500}\right) \log\left(\frac{\frac{1}{2} + \frac{487\tau}{6400}}{z}\right) + \left(\frac{1}{2} - \frac{49\tau}{500}\right) \log\left(\frac{\frac{1}{2} - \frac{487\tau}{6400}}{1 - z + \frac{\tau^2}{512}}\right) > 0 \qquad (6)$$

for all $\tau \in (0, 1)$. The optimization problem corresponding to $z$ can be considered first. Let $g_\tau(z) = \left(\frac{1}{2} + \frac{49\tau}{500}\right) \log\left(\frac{1}{z}\right) + \left(\frac{1}{2} - \frac{49\tau}{500}\right) \log\left(\frac{1}{1 - z + \frac{\tau^2}{512}}\right)$. The derivative of $g_\tau$ w.r.t $z$ equals,

$$\frac{\partial g_\tau(z)}{\partial z} = \frac{\left(\frac{1}{2} + \frac{49\tau}{500}\right)}{z} - \frac{\left(\frac{1}{2} - \frac{49\tau}{500}\right)}{1 - z + \frac{\tau^2}{512}}$$

Thus, this expression has a single minimizer at

$$z^*(\tau) = \frac{\frac{1}{2} + \frac{\tau^2}{1024} + \frac{49\tau^3}{500} + \frac{49\tau^2}{512*500}}{1 - \frac{49\tau}{250}}$$

A simple algebraic substitution shows us that $z^*(\tau) \ge \frac{1}{2} + \frac{\tau}{128}$. Thus the right value to substitute for $z$ in the expression above equals the boundary point $\frac{1}{2} + \frac{\tau}{128}$. Substituting this expression back into the optimization problem 6 it remains to show that for all $\tau \in (0, 1)$,

$$\left(\frac{1}{2} + \frac{49\tau}{500}\right) \log\left(\frac{\frac{1}{2} + \frac{487\tau}{6400}}{\frac{1}{2} + \frac{\tau}{128}}\right) + \left(\frac{1}{2} - \frac{49\tau}{500}\right) \log\left(\frac{\frac{1}{2} - \frac{487\tau}{6400}}{\frac{1}{2} - \frac{\tau}{128} + \frac{\tau^2}{512}}\right) > 0$$

The last expression can be rewritten as,

$$D_{\text{KL}}\left(\frac{1}{2} + \frac{487\tau}{6400} \,\|\, \frac{1}{2} + \frac{\tau}{128}\right) + \underbrace{\left(\frac{49\tau}{500} - \frac{487\tau}{6400}\right)\left(\log\left(\frac{\frac{1}{2} + \frac{487\tau}{6400}}{\frac{1}{2} + \frac{\tau}{128}}\right) - \log\left(\frac{\frac{1}{2} - \frac{487\tau}{6400}}{\frac{1}{2} - \frac{\tau}{128}}\right)\right)}_{\ge 0} +$$
$$\left(\frac{1}{2} - \frac{49\tau}{500}\right) \log\left(\frac{\frac{1}{2} - \frac{\tau}{128}}{\frac{1}{2} - \frac{\tau}{128} + \frac{\tau^2}{512}}\right)$$

By Pinsker's inequality (see Lemma 2),

$$D_{\text{KL}}\left(\frac{1}{2} + \frac{487\tau}{6400} \,\|\, \frac{1}{2} + \frac{\tau}{128}\right) \ge \frac{1}{2\ln(2)}\left(2 * \left(\frac{487}{6400} - \frac{1}{128}\right)\tau\right)^2 \ge 0.013\tau^2.$$

The following inequalities also hold,

$$\frac{\frac{1}{2} - \frac{\tau}{128}}{\frac{1}{2} - \frac{\tau}{128} + \frac{\tau^2}{512}} = 1 - \frac{\frac{\tau^2}{512}}{\frac{1}{2} - \frac{\tau}{128} + \frac{\tau^2}{512}} \geq 1 - \left(\frac{\tau^2}{512}\right)/\left(\frac{1}{2}\right) = 1 - \frac{\tau^2}{256}.$$

Since for all $x \leq \frac{1}{256}$ we have that $g(x) = \log(1-x) + 2x$ is increasing for $0 \leq x \leq \frac{1}{2}$ and sinc $\tau \leq 1$ this implies that

$$\log\left(\frac{\frac{1}{2} - \frac{\tau}{128}}{\frac{1}{2} - \frac{\tau}{128} + \frac{\tau^2}{512}}\right) \geq \log\left(1 - \frac{\tau^2}{256}\right) \geq -\frac{\tau^2}{128}.$$

Therefore,

$$\left(\frac{1}{2} - \frac{49\tau}{500}\right) \log\left(\frac{\frac{1}{2} - \frac{\tau}{128}}{\frac{1}{2} - \frac{\tau}{128} + \frac{\tau^2}{512}}\right) \geq -\left(\frac{1}{2} - \frac{49\tau}{500}\right)\frac{\tau^2}{128} = -\frac{201}{64000}\tau > -0.004\tau^2$$

Therefore,

$$\left(\frac{1}{2} + \frac{49\tau}{500}\right) \log\left(\frac{\frac{1}{2} + \frac{487\tau}{6400}}{\frac{1}{2} + \frac{\tau}{128}}\right) + \left(\frac{1}{2} - \frac{49\tau}{500}\right) \log\left(\frac{\frac{1}{2} - \frac{487\tau}{6400}}{\frac{1}{2} - \frac{\tau}{128} + \frac{\tau^2}{512}}\right) \geq 0.013\tau^2 - 0.004\tau^2 = 0.009\tau^2.$$

Since $\boldsymbol{\theta}_{\text{constant}}$ parametrizes an $L-$Lipschitz function, $f_{\boldsymbol{\theta}_{\text{constant}}}$ this finalizes the result. It implies the constant classifier has a better loss than any classifier having at least one negative label.

$\square$

The reverse version of Lemma 4 also holds.

**Lemma 4** (Reverse version of Lemma 3). *If $\mathcal{D}_t$ satisfies the following properties,*

1. $\forall (\mathbf{x}, y), (\mathbf{x}', y') \in \mathcal{D}_t$ *it holds that* $\|\mathbf{x} - \mathbf{x}'\| \leq \frac{\tau^2}{128L}$.

2. *There exists* $(\tilde{\mathbf{x}}, y) \in \mathcal{D}_t$ *such that* $f_{\boldsymbol{\theta}_\star}(\mathbf{x}) < 0$.

3. $\hat{y} = \frac{1}{|\mathcal{D}_t|} \sum_{(\mathbf{x}, y) \in \mathcal{D}_t} y \leq \frac{1}{4} + \frac{\mu(-\tau)}{2}$.

*Then,*

$$\widehat{\boldsymbol{\theta}}_t = \arg\min_{\theta} \bar{\mathcal{L}}(\boldsymbol{\theta}|\mathcal{D}_t)$$

*Satisfies,* $f_{\widehat{\boldsymbol{\theta}}_t}(\mathbf{x}) < 0$ *for all* $(\mathbf{x}, y) \in \mathcal{D}_t$.

*Proof.* The proof of Lemma 3 applies to this setting. $\square$

We'll use the notation $\mathcal{D}_t(1, R, \mathbf{x}) = \{(\mathbf{x}, y) \in \mathcal{D}_t(R, \mathbf{x}) \text{ s.t. } y = 1\}$ and $\mathcal{D}_t(0, R, \mathbf{x}) = \{(\mathbf{x}, y) \in \mathcal{D}_t(R, \mathbf{x}) \text{ s.t. } y = 0\}$ and $\hat{y}(\mathbf{x}) = \frac{|\mathcal{D}_t(1, \frac{\tau^2}{128L}, \mathbf{x})|}{|\mathcal{D}_t(\frac{\tau^2}{128L}, \mathbf{x})|}$

Let's consider a $\frac{\tau^2}{256L}$-cover $\mathcal{N}(B, \frac{\tau^2}{256L})$ of the radius $B$-ball (for an in depth discussion of properties of $\epsilon-$covers see Chapter 5 of [41] ) in $\mathbb{R}^d$. We will further refine this cover into one made of disjoint subsets. It is easy to see that such a cover can be constructed out of a covering made of possibly overlapping balls via the following steps. We further trim the cover to be made of regions all with positive probability under $\mathcal{P}_\mathcal{X}$.

1. Since $\mathcal{N}(B, \frac{\tau^2}{256L})$ is finite any point $\mathbf{x} \in B(\mathbf{0}, B)$ lies in the intersection of finitely many elements from $\mathcal{N}(B, \frac{\tau^2}{256L})$.

2. For each $n \in |\mathcal{N}(B, \frac{\tau^2}{256L})|$ the subset of points of $B(\mathbf{0}, B)$ that lie in the intersection of exactly $n$ balls from $\mathcal{N}(B, \frac{\tau^2}{256L})$ is a finite collection of connected subsets.

3. For each region alluded in the previous item and within each ball of $\mathcal{N}(B, \frac{\tau^2}{256L})$, assign a specific ball to be the one preserving that region. All of this is possible because these sets are finite.

4. The previous procedure induces the desired disjoint covering.

Let $\tilde{\mathcal{N}}(B, \frac{\tau^2}{256L})$ be that cover. For any $\mathbf{x} \in B(\mathbf{0}, B)$ we will use the notation $s(\mathbf{x})$ to denote the element of $\tilde{\mathcal{N}}(B, \frac{\tau^2}{256L})$ containing $\mathbf{x}$ and $b(\mathbf{x})$ to denote the center of the ball (inherited from the original covering) whose modified version $(s(\mathbf{x}))$ in $\tilde{\mathcal{N}}(B, \frac{\tau^2}{256L})$ contains $\mathbf{x}$.

Let's define a quantized population distribution $\mathcal{P}^b$ over $\left\{ \bar{\mathbf{x}} \text{ s.t. } \bar{\mathbf{x}} \text{ is a 'center'' of an element in } \tilde{\mathcal{N}}(B, \frac{\tau^2}{256L}) \right\} \times \{0, 1\}$ with probabilities $\mathcal{P}^b(\bar{\mathbf{x}}) = \int_{\mathbf{x} \text{ s.t. } b(\mathbf{x})=\bar{\mathbf{x}}} \mathcal{P}_{\mathcal{X}}(\mathbf{x}) d\mathbf{x}$ for $\bar{\mathbf{x}} \in \mathcal{N}(B, \frac{\tau^2}{256L})$. And $\mathcal{P}^b(\bar{y} = 1 | \bar{\mathbf{x}}) = \frac{\int_{\mathbf{x} \text{ s.t. } b(\mathbf{x})=\bar{\mathbf{x}}} \mathcal{P}(y=1, \mathbf{x}) dx}{\mathcal{P}^b(\bar{\mathbf{x}})}$.

For any $x \in B(\mathbf{0}, B)$ we define $\bar{y}(\bar{\mathbf{x}}, R) = \mathcal{P}_{\mathbf{x} \sim \mathcal{P}_{\mathcal{X}}, y \sim \mathrm{Ber}(\mu(f_{\boldsymbol{\theta}_\star}(\mathbf{x})))}(y = 1 | \mathbf{x} \in B(\bar{\mathbf{x}}, R))$ to be the conditional

Let $\mathbf{x} \in \mathbf{supp}(\mathcal{P}_{\mathcal{X}})$ be any point in the support of $\mathcal{P}_{\mathcal{X}}$. By $\bar{\mathbf{x}} = b(\mathbf{x})$ from $\mathcal{N}(B, \frac{\tau^2}{256L})$ satisfies $B(\bar{x}, \frac{\tau^2}{128L}) \subset B(\mathbf{x}, \frac{\tau^2}{128L})$. Consequently at any time $t$, point $\mathbf{x}_t$ satisfies $\{\mathbf{x} \in \mathcal{D}_t \text{ s.t. } \mathbf{x} \in s(\mathbf{x}_t)\} \subseteq \mathcal{D}_t(\frac{\tau^2}{128L}, \mathbf{x}_t)$.

The following concentration result will prove useful,

**Lemma 5** (Hoeffding Inequality). *Let $\{M_t\}_{t=1}^{\infty}$ be a martingale difference sequence with $|M_t| \leq \zeta$ and let $\delta \in (0, 1]$. Then with probability $1 - \delta$ for all $T \in \mathbb{N}$*

$$\sum_{t=1}^{T} M_t \leq 2\zeta \sqrt{T \ln\left(\frac{6 \ln T}{\delta}\right)}.$$

for a proof see Lemma A.1 from [7].

Let $\mathbf{x}$ be a fixed point in $B(\mathbf{0}, B)$. Let's define the martingale sequences $M_t^{(1)}(\mathbf{x}) = \mathcal{P}_{\mathcal{X}}(\tilde{\mathbf{x}} \in B(\mathbf{x}, \frac{\tau}{128L})) - \mathbf{1}\left\{\mathbf{x}_t \in B(\mathbf{x}, \frac{\tau^2}{128L})\right\}$ and $M_t^{(2)}(\mathbf{x}) = \mathbf{1}\left\{\mathbf{x}_t \in B(\mathbf{x}, \frac{\tau^2}{128L})\right\} \cdot \left(y_t - \bar{y}(\mathbf{x}, \frac{\tau^2}{128L})\right)$. As a consequence of Lemma 5 we see that with probability at least $1 - \delta$ for all $t \in \mathbb{N}$,

$$\sum_{t=1}^{T} M_t^{(1)} \leq 4\sqrt{t \ln\left(\frac{6 \ln t}{\delta}\right)} \tag{7}$$

Let's define this event as $\mathcal{E}_1(\delta)$. And similarly with probability at least $1 - 2\delta$ for all $t \in \mathbb{N}$,

$$\left| \sum_{t=1}^{T} M_t^{(2)} \right| \leq 4\sqrt{t \ln\left(\frac{6 \ln t}{\delta}\right)} \tag{8}$$

Let's define this event as $\mathcal{E}_2(\delta)$.

Let $p_{\min} = \min_{s \in \tilde{\mathcal{N}}(B, \frac{\tau^2}{256L})} \mathcal{P}_{\mathcal{X}}(s)$. Equation 7 implies that whenever $\mathcal{E}_1(\delta)$ holds, for all $t \in \mathcal{N}$

$$p_{\min} t \leq t \mathcal{P}_{\mathcal{X}}(\tilde{\mathbf{x}} \in B(\mathbf{x}, \frac{\tau}{128L})) \leq \sum_{\ell=1}^{t} \mathbf{1}\left\{\mathbf{x}_\ell \in B(\mathbf{x}, \frac{\tau^2}{128L})\right\} + 4\sqrt{t \ln\left(\frac{6 \ln t}{\delta}\right)} \tag{9}$$

Let $t_0 \in \mathbb{N}$ be the first integer $t$ such that $p_{\min} t - 4\sqrt{t \ln\left(\frac{6 \ln t}{\delta}\right)} \geq \frac{p_{\min} t}{2}$. For all $t \geq t_0$ we have that

$$\frac{p_{\min}}{2} t \leq \sum_{\ell=1}^{t} \mathbf{1}\left\{\mathbf{x}_\ell \in B\left(\mathbf{x}, \frac{\tau^2}{128L}\right)\right\}$$

We will make use of the following supporting result,

**Lemma 6.** *Let $c_1 \geq 1, c_2 > 0$. For all $t \geq 4c_1 \log(4c_1c_2)$,*

$$t \geq c_1 \log(c_2 t)$$

*Proof.* The following fact will prove useful,

1. The function $x \geq \ln(x)$ for all $x \geq 1$.

    - **Proof:** Let $g(x) = x - \ln(x)$, observe that $g(1) = 0$ and $g'(x) = 1 - \frac{1}{x} \geq 0$ for all $x \geq 1$. This finalizes the proof.

Let's start by expanding $c_1 \log(c_2 t) = c_1 \log(c_2) + c_1 \log(t)$. A necessary condition for the inequality $\frac{t}{2} \geq c_1 \log(c_2 t)$ to hold is that $t \geq c_1 \log(c_2)$. Consider the function $g(t) = \frac{t}{2} - c_1 \log(t)$. It's derivative equals $g'(t) = \frac{1}{2} - \frac{c_1}{t}$ which implies that $g$ is increasing for all $t \geq 2c_1$.

Since $c_1 \geq 1$,

$$\log(4c_1) \geq \log\log(4c_1)$$

Thus

$$4\log(4c_1) \geq 2\log(4c_1) + 2(\log\log(4c_1))$$

And therefore

$$4c_1 \log(4c_1) \geq 2c_1 \log(4c_1 \log(4c_1))$$

this implies that $g(4c_1 \log(4c_1)) \geq 0$. The increasing nature of $g$ for all $t \geq 2c_1$ implies that as long as $t \geq 2c_1 \log(c_2) + 4c_1 \log(4c_1)$, then $t \geq c_1 \log(c_2 t)$. We can relax that condition to $t \geq 4c_1 (\log(c_2) + \log(4c_1))$, the result follows.

$\square$

We can derive a more precise bound for $t_0$ as follows,

**Lemma 7.** *With probability $1 - \delta$, for all $t \geq \frac{256}{p_{\min}^2} \ln\left(\frac{768}{p_{\min}^2 \delta}\right)$ we have that*

$$\frac{p_{\min}}{2} t \leq \sum_{\ell=1}^{t} \mathbf{1}\left\{\mathbf{x}_\ell \in B\left(\mathbf{x}, \frac{\tau^2}{128L}\right)\right\} \tag{10}$$

*Proof.* We only need to show that $t \geq \frac{256}{p_{\min}^2} \ln\left(\frac{768}{p_{\min}^2 \delta}\right)$ is a sufficient choice for $t_0$. Recall $t_0$ is the first integer such that $p_{\min} t - 4\sqrt{t \ln\left(\frac{6 \ln t}{\delta}\right)} \geq \frac{p_{\min} t}{2}$. The following two facts will prove useful,

1. The function $x \geq \ln(x)$ for all $x \geq 1$.

    - **Proof:** Let $g(x) = x - \ln(x)$, observe that $g(1) = 0$ and $g'(x) = 1 - \frac{1}{x} \geq 0$ for all $x \geq 1$. This finalizes the proof.

2. The function $x \geq 2\ln(x)$ for all $x \geq 2$.

    - **Proof:** Let $g(x) = x - \ln(2x)$, observe that $g(2) > 0$ and that $g'(x) = 1 - \frac{1}{x} \geq 0$ for all $x \geq 1$. This finalizes the proof.

The required inequality $p_{\min}t - 4\sqrt{t\ln\left(\frac{6\ln t}{\delta}\right)} \geq \frac{p_{\min}t}{2}$ holds if

$$\frac{p_{\min}}{2}t \geq 4\sqrt{t\ln\left(\frac{6\ln t}{\delta}\right)}$$

Which holds iff,

$$\frac{p_{\min}^2}{64}t \geq \ln\left(\frac{6\ln t}{\delta}\right) = \ln\left(6\ln(t)\right) + \ln\left(\frac{1}{\delta}\right)$$

It is enough to set $t_0$ such that

$$\frac{p_{\min}^2 t}{128} \geq \ln\left(\frac{1}{\delta}\right). \tag{11}$$

and,

$$\frac{p_{\min}^2 t}{128} \geq \ln\left(6\ln(t)\right). \tag{12}$$

Equation 11 yields the requirement

$$t \geq \frac{128}{p_{\min}^2}\ln\left(\frac{1}{\delta}\right) \tag{13}$$

Let's observe that $g(t) = \frac{p_{\min}^2}{128}t - \ln(6\ln(t))$ is increasing whenever $g'(t) = \frac{p_{\min}^2}{128} - \frac{1}{\ln(t)t} \geq 0$. This holds if $\ln(t)t \geq \frac{128}{p_{\min}^2}$ which holds for $t \geq \frac{128}{p_{\min}^2}$.

We will deal with Equation 12 by setting

$$t = \frac{128}{p_{\min}^2}\ln\left(\left(\frac{6\times 128}{p_{\min}^2}\right)^2\right) \tag{14}$$

First observe this satisfies the previous requirement of $t \geq \frac{128}{p_{\min}^2}$. Let's see the last setting indeed works by noting this setting satisfies Inequality 12 because we can show,

$$\begin{aligned}
\ln\left(\left(\frac{6\times 128}{p_{\min}^2}\right)^2\right) &= 2\ln\left(\frac{6\times 128}{p_{\min}^2}\right)\\
&\overset{(i)}{\geq} \ln\left(6\ln\frac{128}{p_{\min}^2}\right) + \ln\left(\frac{6\times 128}{p_{\min}^2}\right)\\
&\overset{(ii)}{\geq} \ln\left(6\ln\frac{128}{p_{\min}^2}\right) + \ln\left(2\ln\left(\frac{6\times 128}{p_{\min}^2}\right)\right)\\
&= \ln\left(6\ln\frac{128}{p_{\min}^2}\ln\left(\left(\frac{6\times 128}{p_{\min}^2}\right)^2\right)\right)
\end{aligned}$$

Inequality $(i)$ holds because as noted above for all $x \geq 1$, $x \geq \ln(x)$ and therefore $\frac{128}{p_{\min}^2} \geq \ln\frac{128}{p_{\min}^2}$ and therefore $\ln\left(\frac{6\times 128}{p_{\min}^2}\right) \geq \ln\left(6\ln\frac{128}{p_{\min}^2}\right)$. Inequality $(ii)$ holds because as noted above for all $x \geq 2$, $x \geq \ln(2x)$ and therefore $\ln\left(\frac{6\times 128}{p_{\min}^2}\right) \geq \ln\left(2\ln\left(\frac{6\times 128}{p_{\min}^2}\right)\right)$.

We conclude the proof by combining the condition from Equations 13 and 14 the condition on $t$ can be written as,

$$t \geq \max\left(\frac{128}{p_{\min}^2}\ln\left(\frac{1}{\delta}\right), \frac{128}{p_{\min}^2}\ln\left(\left(\frac{6\times 128}{p_{\min}^2}\right)^2\right)\right)$$

And therefore, it is enough to set $t \geq \frac{256}{p_{\min}^2} \ln \left( \frac{768}{p_{\min}^2 \delta} \right)$.

$\square$

As a consequence of Lemma 7 we conclude that provided $t$ is sufficiently large $\sum_{\ell=1}^{t} \mathbf{1} \left\{ \mathbf{x}_\ell \in B \left( \mathbf{x}, \frac{\tau^2}{128L} \right) \right\}$ grows at a linear rate with large probability. As a consequence of Equation 8,

$$\left| \sum_{\ell=1}^{t} \mathbf{1} \left\{ \mathbf{x}_\ell \in B \left( \mathbf{x}, \frac{\tau^2}{128L} \right) \right\} \cdot \left( \bar{y} \left( \mathbf{x}, \frac{\tau^2}{128L} \right) - y_\ell \right) \right| \leq 4 \sqrt{t \ln \left( \frac{6 \ln t}{\delta} \right)}$$

thus,

$$\bar{y} \left( \mathbf{x}, \frac{\tau^2}{128L} \right) \sum_{\ell=1}^{t} \mathbf{1} \left\{ \mathbf{x}_\ell \in B \left( \mathbf{x}, \frac{\tau^2}{128L} \right) \right\} - 4 \sqrt{t \ln \left( \frac{6 \ln t}{\delta} \right)} \leq \sum_{\ell=1}^{t} y_\ell \mathbf{1} \left\{ \mathbf{x}_\ell \in B \left( \mathbf{x}, \frac{\tau^2}{128L} \right) \right\}$$

(15)

and

$$\sum_{\ell=1}^{t} y_\ell \mathbf{1} \left\{ \mathbf{x}_\ell \in B \left( \mathbf{x}, \frac{\tau^2}{128L} \right) \right\} \leq \bar{y} \left( \mathbf{x}, \frac{\tau^2}{128L} \right) \sum_{\ell=1}^{t} \mathbf{1} \left\{ \mathbf{x}_\ell \in B \left( \mathbf{x}, \frac{\tau^2}{128L} \right) \right\} + 4 \sqrt{t \ln \left( \frac{6 \ln t}{\delta} \right)}$$

(16)

with probability $1 - 2\delta$ for all $t \geq 1$. Equation 15 implies,

$$\bar{y} \left( \mathbf{x}, \frac{\tau^2}{128L} \right) - \frac{4 \sqrt{t \ln \left( \frac{6 \ln t}{\delta} \right)}}{\sum_{\ell=1}^{t} \mathbf{1} \left\{ \mathbf{x}_\ell \in B \left( \mathbf{x}, \frac{\tau^2}{128L} \right) \right\}} \leq \frac{\sum_{\ell=1}^{t} y_\ell \cdot \mathbf{1} \left\{ \mathbf{x}_\ell \in B \left( \mathbf{x}, \frac{\tau^2}{128L} \right) \right\}}{\sum_{\ell=1}^{t} \mathbf{1} \left\{ \mathbf{x}_\ell \in B \left( \mathbf{x}, \frac{\tau^2}{128L} \right) \right\}}$$

$$= \frac{\left| \mathcal{D}_t \left( 1, \frac{\tau^2}{128L}, \mathbf{x} \right) \right|}{\left| \mathcal{D}_t \left( \frac{\tau^2}{128L}, \mathbf{x} \right) \right|}$$

By Lemma 7 if $\mathcal{E}_1(\delta) \cap \mathcal{E}_2(\delta)$ hold, for all $t \geq \frac{256}{p_{\min}^2} \ln \left( \frac{768}{p_{\min}^2 \delta} \right)$ Equation 10 implies,

$$\mu(\tau) - \frac{8 \sqrt{\ln \left( \frac{6 \ln t}{\delta} \right)}}{p_{\min} \sqrt{t}} \leq \bar{y} \left( \mathbf{x}, \frac{\tau^2}{128L} \right) - \frac{8 \sqrt{\ln \left( \frac{6 \ln t}{\delta} \right)}}{p_{\min} \sqrt{t}}$$

$$\leq \bar{y} \left( \mathbf{x}, \frac{\tau^2}{128L} \right) - \frac{4 \sqrt{t \ln \left( \frac{6 \ln t}{\delta} \right)}}{\sum_{\ell=1}^{t} \mathbf{1} \left\{ \mathbf{x}_\ell \in B \left( \mathbf{x}, \frac{\tau^2}{128L} \right) \right\}}$$

$$\leq \frac{\left| \mathcal{D}_t \left( 1, \frac{\tau^2}{128L}, \mathbf{x} \right) \right|}{\left| \mathcal{D}_t \left( \frac{\tau^2}{128L}, \mathbf{x} \right) \right|}$$

If $\mathbf{x}$ satisfies $f_{\boldsymbol{\theta}_\star}(\mathbf{x}) \geq \tau$ then $\bar{y} \left( \mathbf{x}, \frac{\tau^2}{128L} \right) \geq \mu(\tau)$ and therefore the same logic as in the proof of Lemma 7 implies that if in addition $t \geq \frac{128}{\left( \frac{\mu(\tau)}{4} - \frac{1}{8} \right)^2 p_{\min}^2} \ln \left( \frac{384}{\left( \frac{\mu(\tau)}{4} - \frac{1}{8} \right)^2 p_{\min}^2 \delta} \right)$,

$$\frac{8 \sqrt{\ln \left( \frac{6 \ln t}{\delta} \right)}}{p_{\min} \sqrt{t}} \leq \frac{\mu(\tau)}{4} - \frac{1}{8}$$

And therefore,

$$\frac{1}{8} + \frac{3\mu(\tau)}{4} \leq \frac{|\mathcal{D}_t\left(1, \frac{\tau^2}{128L}, \mathbf{x}\right)|}{|\mathcal{D}_t\left(\frac{\tau^2}{128L}, \mathbf{x}\right)|}$$

Thus,

**Corollary 1.** *If $\mathcal{E}_1(\delta) \cap \mathcal{E}_2(\delta)$ holds and $f_{\boldsymbol{\theta}_\star}(\mathbf{x}) \geq \tau$ then for all $t \geq \frac{256}{\left(\frac{\mu(\tau)}{4} - \frac{1}{8}\right)^2 p_{\min}^2} \ln\left(\frac{768}{\left(\frac{\mu(\tau)}{4} - \frac{1}{8}\right)^2 p_{\min}^2 \delta}\right)$,*

$$\frac{1}{8} + \frac{3\mu(\tau)}{4} \leq \frac{|\mathcal{D}_t\left(1, \frac{\tau^2}{128L}, \mathbf{x}\right)|}{|\mathcal{D}_t\left(\frac{\tau^2}{128L}, \mathbf{x}\right)|}$$

Similarly we can show that Equation 16 implies

$$\frac{|\mathcal{D}_t\left(0, \frac{\tau^2}{128L}, \mathbf{x}\right)|}{|\mathcal{D}_t\left(\frac{\tau^2}{128L}, \mathbf{x}\right)|} = \frac{\sum_{\ell=1}^t y_\ell \mathbf{1}\left\{\mathbf{x}_\ell \in B\left(\mathbf{x}, \frac{\tau^2}{128L}\right)\right\}}{\sum_{\ell=1}^t \mathbf{1}\left\{\mathbf{x}_\ell \in B\left(\mathbf{x}, \frac{\tau^2}{128L}\right)\right\}} \leq \bar{y}\left(\mathbf{x}, \frac{\tau^2}{128L}\right) + \frac{4\sqrt{t \ln\left(\frac{6 \ln t}{\delta}\right)}}{\sum_{\ell=1}^t \mathbf{1}\left\{\mathbf{x}_\ell \in B\left(\mathbf{x}, \frac{\tau^2}{128L}\right)\right\}}$$

Thus whenever $\mathbf{x}$ satisfies $f_{\boldsymbol{\theta}_\star}(\mathbf{x}) \leq -\tau$ then $\bar{y}\left(\mathbf{x}, \frac{\tau^2}{128L}\right) \leq \mu(-\tau)$. By Lemma 7 if $\mathcal{E}_1(\delta) \cap \mathcal{E}_2(\delta)$ holds, for all $t \geq$ Equation 10 implies,

$$\frac{|\mathcal{D}_t\left(0, \frac{\tau^2}{128L}, \mathbf{x}\right)|}{|\mathcal{D}_t\left(\frac{\tau^2}{128L}, \mathbf{x}\right)|} = \frac{\sum_{\ell=1}^t y_\ell \mathbf{1}\left\{\mathbf{x}_\ell \in B\left(\mathbf{x}, \frac{\tau^2}{128L}\right)\right\}}{\sum_{\ell=1}^t \mathbf{1}\left\{\mathbf{x}_\ell \in B\left(\mathbf{x}, \frac{\tau^2}{128L}\right)\right\}}$$
$$\leq \bar{y}\left(\mathbf{x}, \frac{\tau^2}{128L}\right) + \frac{4\sqrt{t \ln\left(\frac{6 \ln t}{\delta}\right)}}{\sum_{\ell=1}^t \mathbf{1}\left\{\mathbf{x}_\ell \in B\left(\mathbf{x}, \frac{\tau^2}{128L}\right)\right\}}$$
$$\leq \mu(-\tau) + \frac{8\sqrt{\ln\left(\frac{6 \ln t}{\delta}\right)}}{p_{\min}\sqrt{t}}$$

Following the same argument as in the derivation leading to Corollary 1, the same logic as in the proof of Lemma 7 implies that if in addition $t \geq \frac{128}{\left(\frac{1}{8} - \frac{\mu(-\tau)}{4}\right)^2 p_{\min}^2} \ln\left(\frac{384}{\left(\frac{1}{8} - \frac{\mu(-\tau)}{4}\right)^2 p_{\min}^2 \delta}\right)$,

$$\frac{8\sqrt{\ln\left(\frac{6 \ln t}{\delta}\right)}}{p_{\min}\sqrt{t}} \leq \frac{1}{8} - \frac{\mu(-\tau)}{4}.$$

And therefore,

$$\frac{|\mathcal{D}_t\left(0, \frac{\tau^2}{128L}, \mathbf{x}\right)|}{|\mathcal{D}_t\left(\frac{\tau^2}{128L}, \mathbf{x}\right)|} \leq \frac{1}{8} + \frac{3\mu(-\tau)}{4}.$$

Thus the following sister corollary to 1 holds,

**Corollary 2.** *If $\mathcal{E}_1(\delta) \cap \mathcal{E}_2(\delta)$ holds and $f_{\boldsymbol{\theta}}(\mathbf{x}) \leq -\tau$ then for all $t \geq \frac{256}{\left(\frac{1}{8} - \frac{\mu(-\tau)}{4}\right)^2 p_{\min}^2} \ln\left(\frac{768}{\left(\frac{1}{8} - \frac{\mu(-\tau)}{4}\right)^2 p_{\min}^2 \delta}\right)$*

$$\frac{|\mathcal{D}_t\left(0, \frac{\tau^2}{128L}, \mathbf{x}\right)|}{|\mathcal{D}_t\left(\frac{\tau^2}{128L}, \mathbf{x}\right)|} \leq \frac{1}{8} + \frac{\mu(-\tau)}{4}.$$

Consider the sample points $\{\mathbf{x}_t, y_t\}_{t=1}^{\infty}$ all produced i.i.d. from distribution $\mathcal{P}$. For any $t \in \mathbb{N}$ consider $\mathcal{U}_t$ the 'leave-point-$t$' process $\{\mathbf{x}_\ell, y_\ell\}_{\ell \neq t}$ with skip $t$ indexing.

We will apply Corollaries 1 and 2 to the $\{\mathcal{U}_t\}_{t=1}^{\infty}$ processes with a value of $\delta_t = \frac{\delta}{t^2}$ to obtain the following result,

**Lemma 8.** *With probability at least $1 - 6\delta$, for all $t \geq \frac{10^6}{\left(\frac{1}{8} - \frac{\mu(-\tau)}{4}\right)^2 p_{\min}^2} \log\left(\frac{233}{\left(\frac{1}{8} - \frac{\mu(-\tau)}{4}\right) p_{\min} \delta}\right)$*

*If $\mathbf{x}_t$ satisfies $f_{\boldsymbol{\theta}_\star}(\mathbf{x}_t) \geq \tau$ then,*

$$\frac{1}{8} + \frac{3\mu(\tau)}{4} \leq \frac{\left|\mathcal{D}_t\left(1, \frac{\tau^2}{128L}, \mathbf{x}_t\right)\right|}{\left|\mathcal{D}_t\left(\frac{\tau^2}{128L}, \mathbf{x}_t\right)\right|}$$

*If $\mathbf{x}_t$ satisfies $f_{\boldsymbol{\theta}}(\mathbf{x}_t) \leq -\tau$ then,*

$$\frac{\left|\mathcal{D}_t\left(0, \frac{\tau^2}{128L}, \mathbf{x}_t\right)\right|}{\left|\mathcal{D}_t\left(\frac{\tau^2}{128L}, \mathbf{x}_t\right)\right|} \leq \frac{1}{8} + \frac{\mu(-\tau)}{4}.$$

*Proof.* Since $\mu(-\tau) + \mu(-\tau) = 1$, as a direct consequence of Corollaries 1 and 2 we see that for any $t$ with probability at least $1 - 3\frac{\delta}{t^2}$, if $t$ is such that $t - 1 \geq \frac{256}{\left(\frac{1}{8} - \frac{\mu(-\tau)}{4}\right)^2 p_{\min}^2} \ln\left(\frac{768t^2}{\left(\frac{1}{8} - \frac{\mu(-\tau)}{4}\right)^2 p_{\min}^2 \delta}\right)$, then if $\mathbf{x}_t$ satisfies $f_{\boldsymbol{\theta}_\star}(\mathbf{x}_t) \geq \tau$,

$$\frac{1}{8} + \frac{3\mu(\tau)}{4} \leq \frac{\left|\mathcal{D}_{t-1}\left(1, \frac{\tau^2}{128L}, \mathbf{x}_t\right)\right|}{\left|\mathcal{D}_{t-1}\left(\frac{\tau^2}{128L}, \mathbf{x}_t\right)\right|}$$

And if $\mathbf{x}_t$ satisfies $f_{\boldsymbol{\theta}}(\mathbf{x}_t) \leq -\tau$ then,

$$\frac{\left|\mathcal{D}_{t-1}\left(0, \frac{\tau^2}{128L}, \mathbf{x}_t\right)\right|}{\left|\mathcal{D}_{t-1}\left(\frac{\tau^2}{128L}, \mathbf{x}_t\right)\right|} \leq \frac{1}{8} + \frac{\mu(-\tau)}{4}.$$

Since $2t \geq t - 1$, the following inequality shows it is enough to provide a condition for $t$ being such that $t \geq \frac{1024}{\left(\frac{1}{8} - \frac{\mu(-\tau)}{4}\right)^2 p_{\min}^2} \ln\left(\frac{768t}{\left(\frac{1}{8} - \frac{\mu(-\tau)}{4}\right) p_{\min} \delta}\right)$,

$$t \geq \frac{1024}{\left(\frac{1}{8} - \frac{\mu(-\tau)}{4}\right)^2 p_{\min}^2} \ln\left(\frac{768t}{\left(\frac{1}{8} - \frac{\mu(-\tau)}{4}\right) p_{\min} \delta}\right)$$

$$\geq 2 \times \frac{256}{\left(\frac{1}{8} - \frac{\mu(-\tau)}{4}\right)^2 p_{\min}^2} \ln\left(\frac{768t^2}{\left(\frac{1}{8} - \frac{\mu(-\tau)}{4}\right)^2 p_{\min}^2 \delta}\right)$$

This is satisfied for all $t$ such that,

$$t \geq \frac{4096}{\left(\frac{1}{8} - \frac{\mu(-\tau)}{4}\right)^2 p_{\min}^2} \log\left(\frac{12582912}{\left(\frac{1}{8} - \frac{\mu(-\tau)}{4}\right)^3 p_{\min}^3 \delta}\right)$$

To simplify this expression we can take $t \geq \frac{10^6}{\left(\frac{1}{8} - \frac{\mu(-\tau)}{4}\right)^2 p_{\min}^2} \log\left(\frac{233}{\left(\frac{1}{8} - \frac{\mu(-\tau)}{4}\right)p_{\min}\delta}\right)$

Taking a union bound over all $t \in \mathbb{N}$ (and thus over all processes $\mathcal{U}_t$) and using Lemma 6 implies that for all $t$ such that $t \geq \frac{10^6}{\left(\frac{1}{8} - \frac{\mu(-\tau)}{4}\right)^2 p_{\min}^2} \log\left(\frac{233}{\left(\frac{1}{8} - \frac{\mu(-\tau)}{4}\right)p_{\min}\delta}\right)$ with probability at least $1 - 6\delta$ if $\mathbf{x}_t$ satisfies $f_{\boldsymbol{\theta}_\star}(\mathbf{x}_t) \geq \tau$,

$$\frac{1}{8} + \frac{3\mu(\tau)}{4} \leq \frac{\left|\mathcal{D}_{t-1}\left(1, \frac{\tau^2}{128L}, \mathbf{x}_t\right)\right|}{\left|\mathcal{D}_{t-1}\left(\frac{\tau^2}{128L}, \mathbf{x}_t\right)\right|}$$

And if $\mathbf{x}_t$ satisfies $f_{\boldsymbol{\theta}}(\mathbf{x}_t) \leq -\tau$ then,

$$\frac{\left|\mathcal{D}_{t-1}\left(0, \frac{\tau^2}{128L}, \mathbf{x}_t\right)\right|}{\left|\mathcal{D}_{t-1}\left(\frac{\tau^2}{128L}, \mathbf{x}_t\right)\right|} \leq \frac{1}{8} + \frac{\mu(-\tau)}{4}.$$

$\square$

Let's call the event alluded by in Lemma 8 as $\mathcal{E}_\star$. Note that whenever $\mathcal{E}_\star$ holds, all the events $\mathcal{E}_1(\frac{\delta}{t^2}) \cap \mathcal{E}_2(\frac{\delta}{t^2})$ also hold for all $t$ (each corresponding to $\mathbf{x}_t$).

In the ensuing discussion we'll condition on $\mathcal{E}_\star$.

We are ready to link these results with those of Lemma 3 to derive guarantees for the PLOT algorithm. We'll use the following notations in the following discussion to simplify the notations,

$$A_t = \sum_{\ell=1}^{t-1} \mathbf{1}\left\{\mathbf{x}_\ell \in B(\mathbf{x}_t, \frac{\tau^2}{128L})\right\}$$

$$B_t = \mathcal{P}_\mathcal{X}(\tilde{\mathbf{x}} \in B(\mathbf{x}_t, \frac{\tau}{128L}))$$

$$C_t = \bar{y}\left(\mathbf{x}_t, \frac{\tau^2}{128L}\right)$$

$$D_t = \sum_{\ell=1}^{t-1} y_\ell \mathbf{1}\left\{\mathbf{x}_\ell \in B\left(\mathbf{x}_t, \frac{\tau^2}{128L}\right)\right\}$$

Recall that as stated in Theorem 1 the number of pseudo-labels we will introduce at time $t$ equals

$$W_t = \max\left(4\sqrt{t \ln\left(\frac{6t^2 \ln t}{\delta'}\right)}, \frac{\left(\frac{\mu(\tau)}{2} + \frac{1}{4}\right)A_t - D_t}{\frac{3}{4} - \frac{\mu(\tau)}{2}}\right)$$

Since $\mathcal{E}_1(\frac{\delta}{t^2}) \cap \mathcal{E}_2(\frac{\delta}{t^2})$ holds, Equation 9 implies

$$t\mathcal{P}_\mathcal{X}(\tilde{\mathbf{x}} \in B(\mathbf{x}_t, \frac{\tau}{128L})) \leq \sum_{\ell=1}^{t-1} \mathbf{1}\left\{\mathbf{x}_\ell \in B(\mathbf{x}_t, \frac{\tau^2}{128L})\right\} + W_t$$

Similarly Equation 15 implies,

$$\bar{y}\left(\mathbf{x}_t, \frac{\tau^2}{128L}\right) \sum_{\ell=1}^{t-1} \mathbf{1}\left\{\mathbf{x}_\ell \in B\left(\mathbf{x}_t, \frac{\tau^2}{128L}\right)\right\} \leq \sum_{\ell=1}^{t-1} y_\ell \mathbf{1}\left\{\mathbf{x}_\ell \in B\left(\mathbf{x}_t, \frac{\tau^2}{128L}\right)\right\} + W_t$$

Let's see that in case $\mathbf{x}_t$ is such that $f_{\boldsymbol{\theta}_\star}(\mathbf{x}_t) \geq \tau$, the empirical average of the pseudo-label augmented dataset (where effectively $\mathbf{x}_t$ has been added $W_t$ times) is always at least $\frac{1}{4} + \frac{\mu(\tau)}{2}$ thus satisfying the conditions of Lemma 3. This will imply that PLOT will accept $\mathbf{x}_t$.

The inequalities above are equivalent to the relationships

$$tB_t \leq A_t + W_t$$
$$CA_t \leq D_t + W_t$$

Recall $C_t \geq \mu(\tau)$. So we will instead use $\tilde{C} = \mu(\tau)$. Notice the pseudo-label augmented label ratio equals $\frac{D_t + W_t}{A_t + W_t}$ and that $\frac{D_t + W_t}{A_t + W_t} \geq \tilde{C} - \underbrace{\left(\frac{\mu(\tau)}{2} - \frac{1}{4}\right)}_{\alpha}$ since

$$\frac{D_t + W_t}{A_t + W_t} \geq \tilde{C} - \underbrace{\left(\frac{\mu(\tau)}{2} - \frac{1}{4}\right)}_{\alpha}$$

Iff

$$D_t + W_t \geq \left(\tilde{C} - \alpha\right)(A_t + W_t) = A_t\tilde{C} + W_t\tilde{C} - \alpha A_t - \alpha W_t.$$

Which holds iff $W_t \geq \frac{A_t\tilde{C} - D_t - \alpha A_t}{1 - \tilde{C} + \alpha}$. Thus we conclude that for such setting of $W_t$, if $\mathcal{E}_\star$ holds, any point $\mathbf{x}_t$ with $f_{\boldsymbol{\theta}_\star}(\mathbf{x}_t) \geq \tau$ will be accepted. Since $W_t$ is explicitly designed to satisfy this condition, we conclude that $\mathbf{x}_t$ will be accepted.

We are left with showing that points $\mathbf{x}_t$ such that $f_{\boldsymbol{\theta}_\star}(\mathbf{x}_t) \leq -\tau$ will not be spuriously accepted too many times.

To show this result we will use the fact that for large $t$ the ratio $\frac{D_t + W_t}{A_t + W_t} \approx \frac{D_t}{A_t}$ and for large $t$ this ratio is roughly equal to $\bar{y}\left(\mathbf{x}_t, \frac{\tau^2}{128L}\right)$, which in turn is at most $\mu(-\tau)$.

Recall $C_t = \bar{y}\left(\mathbf{x}_t, \frac{\tau^2}{128L}\right)$ and let $\alpha_t = \frac{1}{4} - \frac{\bar{y}\left(\mathbf{x}_t, \frac{\tau^2}{128L}\right)}{2}$. Notice $\alpha_t \geq 0$ since by Lipschitsness, all points $\mathbf{x}$ in $B(\mathbf{x}_t, \frac{\tau^2}{128L})$ satisfy $f_{\boldsymbol{\theta}_\star}(\mathbf{x}) < 0$. In fact $\alpha_t \geq \frac{1}{4} - \frac{\mu(-\tau)}{2}$.

Similar to the discussion above we see that $\frac{D + W_t}{A_t + W_t} \leq C_t + \alpha_t$ iff

$$D_t + W_t \leq A_tC_t + A_t\alpha_t + C_tW_t + \alpha_tW_t$$

Which holds if $W_t \leq \frac{A_tC_t + A_t\alpha_t - D_t}{1 - C_t - \alpha_t}$. Whenever this condition is triggered, the reverse version of Lemma 3 (Lemma 4) will hold and therefore the whole batch will be rejected.

Recall that $W_t = \max\left(\frac{A_t\tilde{C} - D_t - \alpha A_t}{1 - \tilde{C} + \alpha}, 4\sqrt{t\ln\left(\frac{6t^2\ln t}{\delta}\right)}\right)$. Since $\frac{A_t\tilde{C} - D_t - \alpha A_t}{1 - \tilde{C} + \alpha} < \frac{A_tC_t + A_t\alpha_t - D_t}{1 - C_t - \alpha_t}$, the condition $W_t \leq \frac{A_tC_t + A_t\alpha_t - D_t}{1 - C_t - \alpha_t}$ holds only when $\sqrt{t\ln\left(\frac{6t^2\ln t}{\delta}\right)} \leq \frac{A_tC_t + A_t\alpha_t - D_t}{1 - C_t - \alpha_t}$. It remains to see this condition starts holding for all $t$ large enough.

By Equations 16, 9, and 15 if $\mathcal{E}_\star$ holds,

$$D_t \leq C_tA_t + 4\sqrt{t\ln\left(\frac{6t^2\ln t}{\delta}\right)}$$

$$tB_t \leq A_t + 4\sqrt{t\ln\left(\frac{6t^2\ln t}{\delta}\right)}$$

$$C_tA_t \leq D_t + 4\sqrt{t\ln\left(\frac{6t^2\ln t}{\delta}\right)}$$

Then,

$$\frac{A_t(C_t + \alpha_t) - D_t}{1 - C_t - \alpha} \geq \frac{A_t\alpha_t - 4\sqrt{t\ln\left(\frac{6t^2\ln t}{\delta}\right)}}{1 - C_t - \alpha_t}$$

$$\geq \frac{t\alpha_t B_t + 4(\alpha_t - 1)\sqrt{t\ln\left(\frac{6t^2\ln t}{\delta}\right)}}{1 - C_t - \alpha_t}$$

For the last expression to be at least as large as $4\sqrt{t\ln\left(\frac{6t^2\ln t}{\delta}\right)}$, it is enough that $t$ satisfies

$$t \geq \frac{16(2 - C_t - 2\alpha_t)^2}{\alpha_t^2 B_t^2}\ln\left(\frac{6t^2\ln t}{\delta}\right)$$

for which it is in turn enough to set,

$$t \geq \frac{64(2 - C_t - 2\alpha_t)^2}{\alpha_t^2 B_t^2}\ln\left(\frac{6t}{\delta}\right)$$

Thus by Lemma 6 this is satisfied for all $t$ such that,

$$t \geq \frac{256(2 - C_t - 2\alpha_t)^2}{\alpha_t^2 B_t^2}\ln\left(\frac{1536(2 - C_t - 2\alpha_t)^2}{\alpha_t^2 B_t^2\delta}\right)$$

Relaxing this via the inequality $B_t \geq p_{\min}$ and $0 \leq 2 - C_t - 2\alpha_t \leq 2$, this condition holds for all $t$ such that (provided $\mathcal{E}_\star$ holds)

$$t \geq \frac{1024}{\alpha_t^2 p_{\min}^2}\ln\left(\frac{6144}{\alpha_t^2 p_{\min}^2\delta}\right)$$

We have concluded that whenever $\mathcal{E}_\star$ holds,

1. for all $t$, if $\mathbf{x}_t$ satisfies $f_{\boldsymbol{\theta}_\star}(\mathbf{x}_t) \geq \tau$, the PLOT Algorithm will accept point $\mathbf{x}_t$.

2. For all $t \geq \frac{1024}{\alpha_t^2 p_{\min}^2}\ln\left(\frac{6144}{\alpha_t^2 p_{\min}^2\delta}\right)$, if $f_{\boldsymbol{\theta}_\star}(\mathbf{x}_t) \leq -\tau$ the PLOT Algorithm will reject $\mathbf{x}_t$.

By observing that regret is only collected when a mistake is made and mistakes are only made when a point $\mathbf{x}_t$ with $f_{\boldsymbol{\theta}_\star}(\mathbf{x}_t) \leq -\tau$ is accepted, incurring in an instantaneous regret of *order* $\alpha_t$. Since for any level of $\alpha_t$ the total number of times such a point could have been accepted by PLOT is upper bounded by $\frac{1024}{\alpha^2 p_{\min}^2}\ln\left(\frac{6144}{\alpha^2 p_{\min}^2\delta}\right)$ with probability at least $1 - 6\delta$ for all $t$, we conclude the regret is upper bounded by,

$$\mathcal{R}(t) \leq \max_{\ell \leq t}\mathcal{O}\left(\frac{1}{\alpha_\ell p_{\min}^2}\ln\left(\frac{1}{\alpha_\ell^2 p_{\min}^2\delta}\right)\right) \leq \mathcal{O}\left(\frac{1}{\alpha p_{\min}^2}\ln\left(\frac{1}{\alpha^2 p_{\min}^2\delta}\right)\right)$$

Since $\alpha$ is of the order of $\tau$ this concludes the proof of Theorem 1.