# OpenReview forum: "Neural Pseudo-Label Optimism for the Bank Loan Problem"
_NeurIPS.cc/2021/Conference — NeurIPS 2021 Poster_

### Official Review · Reviewer_XPXH · 2021-07-16

**Rating:** 7
**Confidence:** 3

**Summary:**

The paper studies class of problems, called bank loan problem (BLP), where the learner only observes whether a customer will repay a loan if the loan is issued to begin with.
To solve this problem, the paper proposes a simple and efficient algorithm, Pseudo-Labels for Optimism (PLOT), which adds an optimistic label to a subset of data points rejected by the model and trains the model on all the data seen so far including the optimistically added points. Empirical evidence shows that PLOT achieves competitive performance in comparison to known storng baselines such as NeuralUCB.

**Main Review:**


Strengths:
---------------

- The proposed algorithm PLOT, for bank loan problem, is a simple and intuitive algorithm to enable optimism in the bank loan problem.

- Theorem 1, provides regret analysis for the linear logistic models.

- Theorem 2, states that under some assumptions, PLOT converges to a classifier with perfect accuracy.


Weakness:
---------------

- Some of the assumptions are hard to justify in the Theorem 2 (see Assumptions 4 and 5 in the appendix). I would appreciate any intuition as to why the authors feel this connectedness assumption would be satisfied in the neural setting.

- Proof of the Theorem 2 in the appendix feels more like sketch than any rigorous mathematical argument.

Questions for Authors:
---------------
- What is Algorithm 3 on line 270?

- Your proof of Theorem 2, requires additional assumptions (4, and 5 listed in the appendix). Is there a reason why these assumptions would be satisfied in a Neural setting?

- The bank loan problem is also related to the Selective Classification / Learning with abstention literature, where a network has an extra option to either predict on a new example or reject/abstain the example (implying that the learner is not confident in its prediction on this new example). In this literature, a very simple and strong baseline is thresholding based on the entropy of the prediction ( entropy of the prediction is something your counterfactual loss anyways incorporates). This could be a simple and strong baseline in this problem as well.

- Can you confirm if the dependence on d in the Theorem 1 is linear or \sqrt{d}? The proof in the appendix seems to suggest otherwise.

Writing Clarity:
---------------
Paper is easy to follow except for some of the typos listed below and the proof of Theorem 2 in the appendix.

Line 57: model bey training..
Line 92: the learner has decided to accept or not datapoint xt....
Line 515 (appendix): lemmaconfidenceboundrussacappendix ..




**Time Spent Reviewing:**

4

---

> ### Author Response · Authors · 2021-08-10
> **Initial Response**
>
> We thank the reviewer for an insightful review!
>
> ### Theorem 2
>
> Our connectedness assumption can be thought of similar to a Lipschitz condition: similar data-points (as measured by Euclidean distance on feature space) have similar labels. As for Assumption 4, if the connected components have a finite lower bounded volume, compactness would imply a finite number of components. It is unlikely a function class could have enough capacity to fit a dataset with infinitely many regions. Assumption 5 is to ensure the function class we are working with is expressive enough to be able to represent all possible classifiers shattering the label partition. This ensures that PLOT is able to explore regions of the space where the labels may be negative but there is not sufficient data to establish it. For example, the setting of linearly separable and bounded data falls squarely into this setting.
>
>
> ### Additional Baseline: Learning with abstention
>
> Thank you for the pointer to the learning with abstention literature.
> We’ve reviewed both (Online Learning with Abstention) and (Learning with Rejection), and while we found both to be quite insightful, these problem settings differ significantly from the bank loan problem (BLP) we are considering. The key challenge of the BLP is that rejecting a datapoint (i.e. classifying as negative) means that the true label for this datapoint will not be revealed. This dynamic is entirely missing from the “Learning with Rejection” framework.
>
> Any further pointers to specific relevant papers are much appreciated.
>
>
> In regards to the suggested baseline, we believe this amounts to the Greedy method included in the manuscript. As we are operating with a single-output, fully connected NN, thresholding entropy amounts to thresholding prediction distance from 0.5, which is the implementation of Greedy. A focus on the multi-output setting would certainly be interesting, but is not the core focus of this paper. Please let us know if we are misunderstanding the suggested baseline and its adaptation to the single output setting!
>
>
> ### Questions
>
> **Algorithm 3**
> This is a typo; line 270 should only refer to algorithm 2 (PLOT).
>
> **Theorem 1**
> The dependence on d in Theorem 1 is d^2. Apologies for the typo, we will correct this in the final version!

---

> > ### Comment · Reviewer_XPXH · 2021-08-31
> > **Response to Author Rebuttal**
> >
> > Thank you for clarifying the validity of the assumptions in Theorem 2.
> >
> > As far as the greedy baseline is concerned, I believe it would be one of the simple form of selective classification or learning with abstention schemes you can apply.
> >
> > Here are some of the relevant "Learning with abstention" works
> > - Online Learning with Abstention (https://arxiv.org/pdf/1703.03478.pdf)
> > - Selective Classification with One-Sided Prediction (https://arxiv.org/pdf/2010.07853.pdf)
> > - SelectiveNet : A Deep Neural Network with an Integrated Reject Option (https://arxiv.org/pdf/1901.09192.pdf)
> > - Deep Gamblers: Learning to Abstain with Portfolio Theory (https://arxiv.org/abs/1907.00208)
> >
> > However I do agree that the dynamics as you mention would only appear in the online setting not in the batch setting. But some of these schemes can be adjusted to the problem you are tackling in this paper.
> >
> > Regarding typo in theorem 1, please fix it in the updated version. Your rebuttal says its d^2, the appendix suggest  \sqrt{d} while the main text suggests linear in d. I believe its \sqrt(d) by looking at the proof, if not please clarify this issue.
> >
> > In the light of the author rebuttal and other reviewer comments, I believe the proposed scheme merits acceptance and I'm increasing my score to 7.

---

### Official Review · Reviewer_aBf3 · 2021-07-18

**Rating:** 6
**Confidence:** 4

**Summary:**

This paper first motivates the "bank loan problem", which is a scenario where the learner only observes whether a customer will repay a loan if the loan is issued to begin with. The paper moves on to discussions of this problem and introduces contextual linear bandit formulations. This is extended to the case where the model is a deep neural network, and a Theorem shows theoretical guarantees under the case where data is separable. Experiments show improvement over baseline methods, including the recent NeuralUCB method.

**Limitations And Societal Impact:**

yes

**Main Review:**

### Strengths

- The paper focuses on an important problem characterized as the "bank loan problem" and provides a simple way to use neural networks for this problem.
- Experiments show improvement over baselines.
- Code is provided in the supplementary materials and should be helpful for the research community to reproduce results.

### Weaknesses and questions

- Some of the equations seem to be weird. For example, in the equations after line 225 and 232, the first term has $\log f(x)$, but should this be $\log \mu (f(x))$ since $f:\mathbb{R}^d \rightarrow \mathbb{R}$?
- Notations are confusing. $\mathcal{L}^\lambda(\bf{\theta}|\mathcal{D}_t)$ are used as the ordinary cross entropy loss in Appendix C but is used as the negative cross entropy loss after line 225.
- Line 271 points to Appendix E but I couldn't find Appendix E in the supplementary material.
- In the code, I can see that the neural network is a fully-connected 1 hidden layer neural network. There may be a space issue, but I believe this is basic info and should be included in the main paper.
- In Theorem 1: Is $1-2\delta$ a valid probability when $\delta \in (0,1)$?

#### Minor suggestions
- It would be interesting to see experiments where the separability is violated and how it affects the results.
- line 277: Should $R^d$ be $\mathbb{R}^d$?
- Eq. after line 161: Are the parenthesis located in wrong places?

---

Thank you for the response. I found Appendix C corresponds to the proof of Theorem 2, instead of Appendix E. The author responses in the other reviews about the details of this proof was also helpful. Since my concerns were addressed, I would like to raise my score to 6.

**Time Spent Reviewing:**

4

---

> ### Author Response · Authors · 2021-08-10
> **Initial Response**
>
> Thanks for your review. With regards to equation typos and formatting, we will make the suggested fixes for the final revision. With regards to separability, all of our experimental datasets violate the separability assumption, to varying degrees. This can be assessed via performance of a classifier trained on all data points, rather than those collected under a online learning algorithm.
>
> *Classification Accuracy*
>
> Adult: 85.8%      Table 1 from [1]
>
> Bank: 90.6%      Table 2 from [1]
>
> MNIST: ~96% with FC NN, from [2]
>
>
> *Citations*
>
> [1] Wasserstein Fair Classification. Ray Jiang, Aldo Pacchiano, Tom Stepleton, Heinrich Jiang, Silvia Chiappa.
> [2] http://yann.lecun.com/exdb/mnist/

---

### Official Review · Reviewer_9rUK · 2021-07-18

**Rating:** 7
**Confidence:** 3

**Summary:**

Authors propose neural pseudo-label optimism for solving the bank loan problem. This is an interesting setting and I judge the work in the paper (theory & experiments) to be mostly correct. I consider this paper to be sufficiently novel, although there are certainly a large number of papers on the bank loan problem. I am not an expert on the bank loan problem in particular, so I hope another review can provide more clarity as to this paper’s novelty. I am tentatively voting for a rating of 7, but I would like the authors to address some of my concerns below.

**Limitations And Societal Impact:**

Sure

**Main Review:**

Pros:
- Interesting topic & sufficient novelty (as far as I am aware)
- Complementary experiments & theory
- Acceptable writing quality

Cons:
- Some parts of the theory are slightly contrived (but this could be fixed)
- Experiments could use additional details and polish

I strongly suggest the authors rewrite the “neural realizability” aspect of their theoretical framework (or at least provide more details). I understand the motivation behind it, but it seems like a bit of a stretch. For example, consider Assumption 2. Based on the universal approximation property of DNNs, I don’t doubt Assumption 2, but in practice we have no way of knowing whether or not the architecture of choice actually meets this Assumption (at least not before training on a similar dataset). The authors could go a similar route using something like NTK theory which is the limit of wide-networks and is theoretically sound in terms of training and generalization. I understand there is work show a wider class of overparameterized networks can interpolate data due to local convexity, but still this thread of research is not mentioned either.

Related to the above, I am confused if Theorem 2 relies on the assumption of a convex loss function or also convexity of the loss with respect to the model parameters. The authors should clarify these details or at least include a reference to the supplement in which this is discussed.

EXPERIMENTS:
 - Most details are present but the description could use a bit more clarity. The code is provided in the supplement. I did not attempt to run it.
- Need more details concerning the DNN architectures (unless these are in the supplement and I just missed it)
- For the Bank dataset (which is the most interesting for this topic) does not seem to have major improvement between the proposed method and the baselines. The authors might rescale the y-axis so that we can see the differentiation between the curves for the bank dataset
- Error bars/regions are barely visible
- 3 datasets are ok but MNIST in particular does not seem like a great candidate for this topic. I like the use of “Adult” and “Bank” and encourage the authors to move MNIST to the supplement in the place of 1 or 2 more “realistic” datasets.


Minor:
-typo in line 57: “bey”


**Time Spent Reviewing:**

2.5

---

> ### Author Response · Authors · 2021-08-10
> **RE Official Review of Paper11000 by Reviewer 9rUK**
>
> **Assumption 2 - Neural realizability**
>
> Thanks so much for the reviewers comments.
> Both our neural realizability assumption and NTK are  subject to the same issue: real data may not satisfy any or some of the assumptions of the model of choice. To maximize the likelihood of these modeling assumptions, it is standard to make use of existing model selection approaches (see for example works like “Model Selection for Stochastic Contextual Bandit Problems” [1]). These methods seamlessly combine an ensemble of progressively deeper neural networks and enjoy the regret guarantees of the best approximating neural network model. This makes the neural realizability assumption much more likely.
> Additionally, our algorithm does not require the neural realizability assumption to be used in practice. In fact all of our experiments are conducted on public datasets for which such assumption is not guaranteed to hold, at least not for the type of models that we use in the experimental section.
>
> The latter result is a simple corollary of existing model selection algorithms such as Stochastic Corral [1] used with multiple instances of PLOT based on different neural network architectures. We will add a comment about this in the camera ready version.
>
>
> **Theorem 2**
>
> In this theorem we do not rely on convexity assumptions. Instead, the proof relies on two conditions: separability (described in the proof body), and makes use of the separability as stated in the proof statement as well as a label space shattering condition.
>
> We apologize for a labeling issue in our submission, which directs the reader to Appendix E for further detail.The latter condition is in fact described The proof details are in Appendix C, along with the full details of this theorem.
>
>
> **Experimental Results**
>
> *With regards to presentation:*
>
> We realize that the graph scaling can make the performance improvement on Bank difficult to see, and will provide a table with regret values in the Camera Ready.
>
> The DNN architecture is a single-layer fully connected network, using ADAM for optimization. We will add this description of the DNN to the paper, following the condensation suggested to **Reviewer 7avZ**.
>
>
> *With regards to dataset selection:*
>
> Although we agree that the UCI datasets best represent the theme of the Bank-Loan problem, we do think MNIST is important to include. Performance with high-dimensional function approximation is a key motivation for PLOT. MNIST is by far the highest-dimensional input feature set (d=784), more so than any dataset in similar Deep Bandit papers (e.g. https://arxiv.org/pdf/1802.09127.pdf).
>
> The point on the bank results is well taken. We will either rescale the axes or provide a table format in the camera ready, to make the result clearer. Additionally, we are running additional experiments on the UCI German credit dataset (https://archive.ics.uci.edu/ml/datasets/statlog+(german+credit+data)), to include in the camera ready version. We will add these to our response here ASAP.
>
> *Citations*
>
> [1] Model Selection in Contextual Stochastic Bandit Problems. Aldo Pacchiano, My Phan, Yasin Abbassi-Yadkori, Anup Rao, Julian Zimmert, Tor Lattimore, Csaba Szepesvari.

---

> > ### Comment · Reviewer_9rUK · 2021-08-25
> > **Reviewer response**
> >
> > Thank you for the reply. I find the reply sufficient for me to maintain my score.

---

### Official Review · Reviewer_7avZ · 2021-07-19

**Rating:** 4
**Confidence:** 2

**Summary:**

This paper proposes a new approach to improve the performance of non-linear predictors (DNN models) on binary labels when the observations are endogenously selected. The issue arising in traditional setups is that the algorithm will only select observations with high positive outcome (reward). This may lead to sub-optimal and unfair models generating a large proportion of false negatives. The solution proposed in this paper is to add optimistic labels (with positive rewards) to a subset of points that the model would normally reject. Doing so optimistically bias the model, and stimulate the exploration of the more uncertain regions. The suggested algorithm and paper focus is on online learning methods. Several experiments highlight the competitiveness of this approach compared to similar ones.

**Limitations And Societal Impact:**

* The approach limitations regarding the data independence and separability should be further discussed. Observations and labels in a bank-loan problem would typically be exposed to correlated and time-varying factors depending, for example, on the current health of the economy.

**Main Review:**

* The main contribution of the paper appears to be described in Section 5, which is on pages 6-7. The previous three sections (setting, related work, and background) occupy a lot of space without necessary conveying crucial information about this work. The paper would benefit from some restructuring.
* There is a large literature on self-selection bias beyond the online learning literature which is nowhere discussed.
* Being familiar but not a specialist in online learning (OL), I have the following questions about the approach:
  * OL is typically applied to problems with high data velocity, when model re-training should be made online because a batch approach would be too costly and/or slow. The examples discussed do not convey the necessity of online updates. Similarly, there is no adversarial game-like sequential interaction. Why would one prefer OL as opposed to, for example, an offline approach that would look at the entire historical sample to quantify model uncertainty and then setup an exploration strategy for the next $n$ observations.
  * It seems that the DNN is fully recalibrated after a new data batch is created (e.g. Algorithm 2). Please comment.
* The numerical experiments in section 6 illustrate that the approach is in line with the selected OL benchmarks, but do not highlight a noticeable performance gain or computational advantage.

**Time Spent Reviewing:**

3h

---

> ### Author Response · Authors · 2021-08-10
> **Initial Response**
>
> We thank the reviewer for an insightful review, especially regarding modeling assumptions. We agree that the paper would benefit from more description of PLOTs assumptions on data generation and sampling. Below, we further clarify those assumptions, and suggest a shortening of the paper background to accommodate these important clarifications in the final version.
>
> We also clarify the online learning setting PLOT is concerned with, and expand upon our model performance gains in this setting.
>
> ## Modeling Assumptions
>
> ### IID Assumptions
>
> As noted by the reviewer, there certainly can be correlated and time-varying factors in the bank-loan problem.
>
> PLOT is designed to work with adversarially generated contexts, i.e.when the observations are not i.i.d., or their distributions ($\mathcal{P}(X)$) are changing adversarially, but the conditional distribution of the responses,  $\mathcal{P}(y|X)$, is fixed. In contrast, a naive bandit algorithm (such as epsilon greedy with time dependent exploration) can be driven to incur a high number of false negatives in problems with a changing $\mathcal{P}_t(X)$ schedule.
>
> However, like many bandit algorithms, PLOT does require a stationary reward function (mapping from features to rewards). Relaxing this assumption changes the problem to an adversarial bandit, outside the scope of this work.
>
> ### Separability Assumptions
>
> The theoretical guarantees for PLOT assume separability.
>
> When the data is separable and the function class shatters the label space, adding positive pseudo-labels to any new point belonging to one of the yet unexplored connected components will encourage exploration of any new points arriving from that part of the space. When sufficient points are present, the effect of the pseudo-label in encouraging optimistic exploration over these connected components fades. These assumptions allow us to characterize in a simple and elegant manner how PLOT balances its exploration and exploitation objectives. There is nothing preventing PLOT from being applied to settings where this assumption is violated. In fact, our experimental results are conducted on public datasets for which the separability assumption does not hold.
>
> ### Paper Structure
>
> To incorporate the above details, we have made the following changes to our paper structure, which we will present for the final revision.
>
>
> 1) Reduce the space dedicated to the linear model results, which can be condensed.
>
> 2) Condensed introduction paragraph
>
> 3) Description of our core IID assumptions, as sketched above
>
> 4) Description of separability assumptions, as sketched above
>
> 5) More thorough description of NN architecture
>
> We are also happy to add references to the self-selection bias literature, as mentioned by the reviewer, and welcome any suggestions.
>
> ## Setting and Performance
>
> ### Online learning
>
> Our setting closely mirrors that of the bandit literature. The key feature defining the problem as online is that each point is associated with a timestep, and not available for training before that timestep. This is why we cannot train on the entire dataset a priori, not for computational reasons. As a result, our method does not focus on computational speed-ups.
>
> We do note that, similar to many bandit algorithms, our method can predict/update on an arbitrary set of data. There is no algorithmic distinction between receiving a batch of n points vs 1 point in PLOT, and thus we focus on the single point case without loss of generality.
>
> ### Performance
>
> We think our results here are quite strong, particularly when considering the variance of the regret. We will add a table in the camera ready to make our improvements more clear.
> On MNIST, the variance of NeurUCB is >10x that of PLOT, while PLOT obtains 50% less regret.
> On Adult, we report a 15% improvement in regret, again with lower variance.

---

> > ### Author Response · Authors · 2021-08-31
> > **Follow up**
> >
> > Dear Reviewer 7avZ,
> >
> > We would like to make sure that our response answered your questions. We are particularly interested to see if our explanation about 'Online Learning' helped clarify some of the misunderstanding about our setup.
> >
> > Thanks a lot,
> >
> > The Authors

---

### Author Response · Authors · 2021-08-26
**Colab Demo Link**

Hello,

We noticed the Colab Demo link that was present in our submission is broken. Please access the (anonymized) demo through this link instead

https://colab.research.google.com/drive/1kjukVierl8g-fpmvrCJ2yNI6Yog3qGO5?usp=sharing

In this colab you will be able to

1) Try out PLOT with a variety of datasets as present in the paper, thus being able to reproduce the experiments present in the paper.
2) Try out and compare vs other baseline algorithms such as $\epsilon$-greedy and neural UCB.
3) View the evolution of the PLOT decision boundary on a simple 2 dimensional dataset illustrating how Pseudo-label optimism aids in exploration.

We hope this tool can help address some of the reviewer's concerns.

Thanks a lot,

The Authors

---

### Decision · Program_Chairs · 2021-09-27

**Decision:**

Accept (Poster)

**Comment:**

The scores from four reviews are 4, 5, 5, and 7. The authors have a persuasive reply to the review with the lowest score, so this score is too harsh. As the authors say, the criticisms in one of the reviews with a 5 score are rather superficial. Overall, the paper is on an important topic and is correct and novel, so I lean towards publication.

One issue not mentioned in the paper is delayed feedback. When a bank grants a loan, a lot of time passes before the true label becomes available, and during this time the function to be learned p(y|x) may change. A related issue mentioned by a reviewer is correlation between labels of different examples, as caused for example by a recession. The last section should discuss these issues briefly, and also discuss fairness, as opposed to merely mentioning it. Lack of exploration can lead to never giving loans to members of a group that has historically faced prejudice. Of course, the three issues just mentioned apply more broadly than just specifically to granting of credit.